# Piezo1 as a force-through-membrane sensor in red blood cells

George Vaisey[1], Priyam Banerjee[2], Alison J North[2], Christoph A Haselwandter[3], Roderick MacKinnon[1]*

[1]Laboratory of Molecular Neurobiology and Biophysics, Howard Hughes Medical Institute, The Rockefeller University, New York, United States; [2]Bio-Imaging Resource Center, The Rockefeller University, New York, United States; [3]Department of Physics and Astronomy and Department of Quantitative and Computational Biology, University of Southern California, Los Angeles, United States

**Abstract** Piezo1 is the stretch activated $Ca^{2+}$ channel in red blood cells that mediates homeostatic volume control. Here, we study the organization of Piezo1 in red blood cells using a combination of super-resolution microscopy techniques and electron microscopy. Piezo1 adopts a non-uniform distribution on the red blood cell surface, with a bias toward the biconcave 'dimple'. Trajectories of diffusing Piezo1 molecules, which exhibit confined Brownian diffusion on short timescales and hopping on long timescales, also reflect a bias toward the dimple. This bias can be explained by 'curvature coupling' between the intrinsic curvature of the Piezo dome and the curvature of the red blood cell membrane. Piezo1 does not form clusters with itself, nor does it colocalize with F-actin, Spectrin, or the Gardos channel. Thus, Piezo1 exhibits the properties of a force-through-membrane sensor of curvature and lateral tension in the red blood cell.

## Editor's evaluation

This study reveals the super-resolution localization of native Piezo1 channels in red blood cells. The data provide further evidence to support the force-through membrane mechanism of mechanotransduction. Notably, the authors find that the distribution of proteins on a cell surface can be governed by membrane curvature.

*For correspondence:
mackinn@rockefeller.edu

**Competing interest:** The authors declare that no competing interests exist.

## Introduction

Red blood cells (RBCs) experience significant mechanical forces in their lifetime as they traverse the mammalian circulatory system, squeezing through capillaries of smaller diameters than themselves (*Chien, 1987*). Central to the ability of RBCs to deform in response to shear stress is their biconcave disk shape (*Lister, 1827*; *Mohandas and Evans, 1994*). RBCs lack organelles and a transcytosolic network and their biconcave shape is instead imparted by the plasma membrane and the membrane skeleton, a 2D quasihexagonal network of actin filament (F-actin) nodes connected by $(\alpha_1\beta_1)_2$-spectrin tetramers bound to transmembrane proteins (*Fowler, 2013*; *Gratzer, 1981*). In response to shear stress a $Ca^{2+}$ influx is activated in RBCs (*Larsen et al., 1981*; *Dyrda et al., 2010*). $Ca^{2+}$ entry into RBCs modulates junctional protein interactions in the membrane skeleton (*Nunomura, 2006*) in addition to activating the Gardos channel (*Gardos, 1958*), a $K^+$ channel, leading to $K^+$ efflux and subsequent cell volume decrease (*Danielczok et al., 2017*), which together facilitate flexibility of the cell membrane.

Piezo1 was first suggested as the mechanosensitive $Ca^{2+}$ channel in RBCs after identification of gain-of-function mutations in patients with dehydrated xerocytosis (*Bae et al., 2013*; *Albuisson et al., 2013*; *Andolfo et al., 2013*; *Zarychanski et al., 2012*). Subsequent studies using a hematopoietic

PIEZO1 knockout mouse confirmed that the channel is responsible for stretch-induced calcium influx in RBCs (*Cahalan et al., 2015*). How Piezo1 is organized in the RBC membrane, however, is not known. The unique biconcave shape of the RBC poses interesting questions about how membrane curvature, which locally is known to be important for Piezo1 structure (*Guo and MacKinnon, 2017*; *Lin et al., 2019*; *Haselwandter and MacKinnon, 2018*), might globally impact Piezo1 distribution and function. It has been proposed that due to its intrinsic curvature, Piezo1 might tend to concentrate in the dimple region of RBCs (*Svetina et al., 2019*), but direct experimental evidence is lacking. More broadly, how Piezo channels are organized in cell membranes is poorly understood. Based on electrophysiological recordings in overexpressing cells (*Gottlieb et al., 2012*) or fluorescence microscopy of a stable cell line (*Ridone et al., 2020*), some authors have reported clustering of Piezo1, but channel densities in both cases are likely non-physiological. Extensive patch-clamp studies and simulation of Piezo1 densities in neuro2Acells, which express Piezo1 natively, led others to the conclusion that Piezo1 channels are homogenously expressed across cell membranes and function as independent mechanotransducers (*Lewis and Grandl, 2021*). Additionally, it is unclear what role the actin cytoskeleton plays in organizing and gating the Piezo1 channel in cells. Piezo1 can be pressure activated in membrane blebs lacking cytoskeleton (*Cox et al., 2016*) and exhibits spontaneous openings in asymmetric lipid droplet bilayers (*Syeda et al., 2016*), consistent with a 'force-from-lipids' model of activation (*Martinac et al., 1990*). Others claim that Piezo1 is tethered to the actin cytoskeleton and is gated by a 'force-from-filament' mechanism (*Wang et al., 2022*).

Here, we set out to image and analyze the organization of Piezo1 channels in RBC membranes. The small size of RBCs and their inherent autofluorescence makes them refractory to conventional fluorescence microscopy, so we employed more sensitive super-resolution microscopic methods. Fluorescent Piezo1 spots can be resolved into single Piezo1 channels which, along with electron microscopy studies, confirm that they do not cluster in RBC membranes. Although not clustering, we

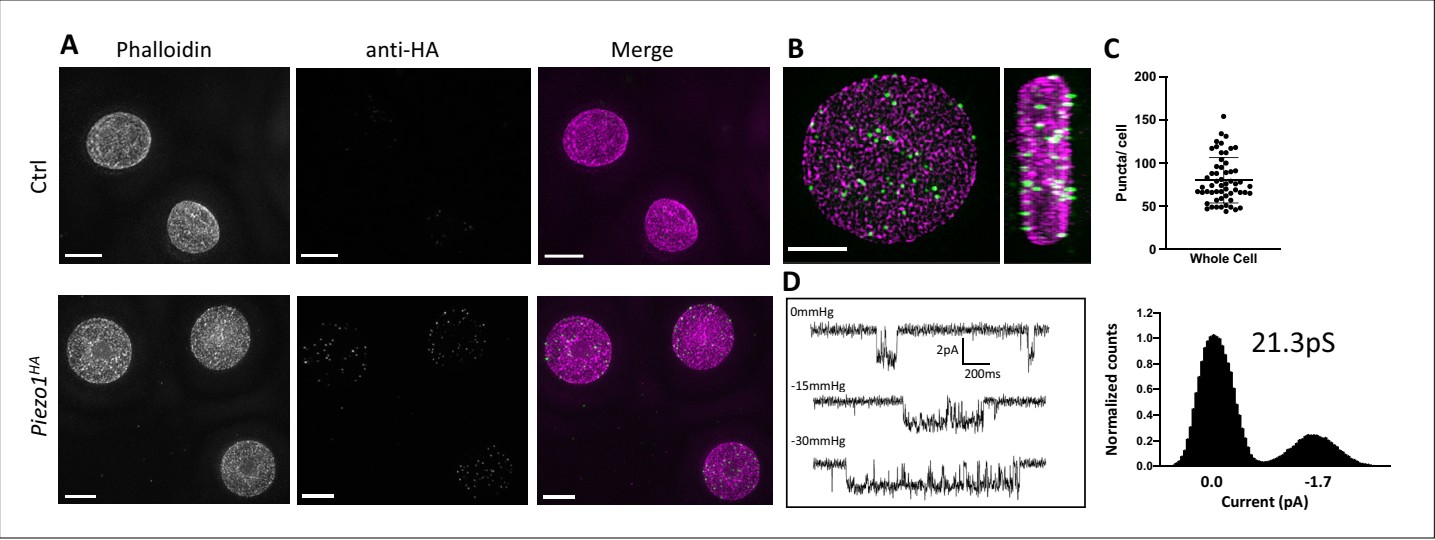

**Figure 1.** Detection of Piezo1 in RBCs. (**A**) Reconstructed 3D-structured illumination microscopy (3D-SIM) images of red blood cells (RBCs), shown as maximum projections of z-stacks, from control (Ctrl, top row) or *Piezo1^HA^* (bottom row) mice labeled using rhodamine phalloidin (magenta) and an antibody directed against the HA tag of Piezo1 (green). Scale bar, 3 μm. (**B**) 3D volume rendering of a representative labeled Piezo1-HA-KI mouse RBC displayed as *XY* (left) and *XZ* (right) axis views, showing the distribution of Piezo1. Scale bar, 2 μm. (**C**) Quantification of the number of Piezo1 spots per cell from 3D-SIM maximum projections. Mean ± standard deviation (SD) = 80.2 ± 26 spots per cell. Numbers were calculated on individual cells (*n* = 57) using Imaris spot detection. (**D**) Representative single channel traces from excised inside-out patch recordings of RBCs in symmetric 150 mM NaMethanesulfonate. The patch was clamped at −80 mV and negative pressure (0 to −30 mm Hg) was applied to the patch pipette. Amplitude histogram (right) enables estimation of single channel conductance. Showing one trace from *n* = 4 recordings.

The online version of this article includes the following figure supplement(s) for figure 1:

**Figure supplement 1.** In HEK293 cells HA-tagged Piezo1 behaves like wildtype Piezo1 in electrophysiology recordings and can be specifically detected by immunostaining.

**Figure supplement 2.** Representative images of *XY* and *XZ* two dimensional slice views from SIM images of immunostained red blood cells (RBCs) from Piezo1-HA mice.

find a non-uniform distribution of Piezo1 in the RBC membrane with enrichment at the dimple, which can be rationalized through the energetic coupling of the intrinsic curvature of Piezo1 to the curvature of the surrounding membrane. Single particle tracking (SPT) studies of Piezo1 demonstrate its lateral mobility in the plasma membrane. This is consistent with co-immunostaining analysis, suggesting that Piezo1 is not tethered to the membrane skeleton. Mobility of Piezo1 in the RBC membrane permits Piezo1 to explore the curvature landscape of the RBC surface and may allow Piezo1 to respond to local and global changes in membrane curvature as well as membrane tension.

## Results

### Expression and localization of Piezo1 in red blood cells

To detect Piezo1 in RBCs a *Piezo1*[HA] mouse line was generated (Jackson Laboratories) in which an HA epitope tag was inserted into an extracellular loop of the PIEZO1 gene. The HA-tagged Piezo1 channel functions similar to the WT channel in electrophysiological recordings of HEK cells and could be specifically detected by immunostaining (*Figure 1—figure supplement 1*). Importantly, only RBCs from the *Piezo1*[HA] mice and not wildtype mice showed fluorescent signal when immunostained with an anti-HA antibody (*Figure 1A*), confirming clear expression of Piezo1 in RBCs. As shown in 3D reconstructions, single optical sections and volume renderings from reconstructed 3D-structured illumination microscopy (3D-SIM) images (*Gustafsson, 2000*; *Heintzmann and Huser, 2017*) of immunostained RBCs, Piezo1 channels appear as spots (*Figure 1A, B* and *Figure 1—figure supplement 2*). These spots are distributed on the RBC membrane, here contoured by phalloidin staining of actin, which forms part of the actin–spectrin network bound to the cytosolic face of the RBC membrane (*Ballas and Krasnow, 1980*; *Figure 1—figure supplement 2*). 3D reconstructions from z-stacks reveal ~80 Piezo1 puncta per RBC (*Figure 1C*), suggesting a relatively low copy number of channels per RBC, similar to estimations of Gardos channel number per RBC (*Lew et al., 1982*; *Brugnara et al., 1993*) and in contrast to very abundant RBC membrane proteins such as Band3, which is estimated to have a copy number of 1 million per cell (*Fairbanks et al., 1971*). Supportively, our patch-clamp recordings of RBCs occasionally identified single channel currents that are stretch activated and have a conductance consistent with other reports of Piezo1 (*Coste et al., 2010*; *Del Mármol et al., 2018*; *Harraz et al., 2022*; *Figure 1D*).

### Piezo1 does not cluster in RBC membranes

Whether Piezo1 channels cluster in membranes to form mechanosensory domains or are distributed more randomly as independent mechanosensors remains unknown. In 3D-SIM images of labeled intact RBCs, fluorescent Piezo1 spots are relatively uniform in size and do not appear to cluster (*Figure 1*). However, our SIM data are diffraction limited by an *XY* resolution of ~120 nm at best and thus each individual spot could feasibly contain a few Piezo1 channels (~23 nm diameter) tightly packed together. To address whether fluorescent spots represent single or multiple Piezo1 channels we therefore employed stimulated emission depletion (STED) microscopy (*Hell and Wichmann, 1994*), which can achieve resolutions below 50 nm and down to for example 22 nm after deconvolution (*Schoonderwoert et al., 2013*), depending on the selected dyes. Owing to the absorption of high-power lasers by heme (*Schloetel et al., 2019*), it was not possible to image intact RBCs by STED. Instead, RBCs adhered to poly-lysine coated coverslips were unroofed by a stream of isotonic buffer (*Swihart et al., 2001*), washing away hemoglobin and leaving patches of RBC membrane still bound to the actin–spectrin network (*Figure 2A*). The approximate flat dimensionality of unroofed RBCs also enabled us to use facilitated image acquisition 2D STED mode to achieve maximal *XY* resolutions of ~40 nm by full width half maximum (FWHM) in single images, closer to the size of a Piezo1 channel. Approximately 26 ± 7 (*n* = 12 cells) spots were observed per unroofed RBC, which equates to ~0.6 puncta/μm$^2$, consistent with the Piezo1 abundance observed in SIM images of intact RBCs. Piezo1 spots in 3D-SIM images of intact RBCs show a range of nearest neighbor distances (*Figure 2B*) with an average nearest spot distance of 540 ± 37 nm (*n* = 19 cells). This is similar to the observed distribution of Piezo1 spots in 2D STED images of unroofed RBC membranes, where the nearest spot distance is 544 ± 106 nm (*n* = 12 cells) and is consistent with fluorescent Piezo1 spots not clustering in the RBC membrane. Further cluster analysis of Piezo1 spots in 2D STED images by 2D spatial analysis (*Andrey*

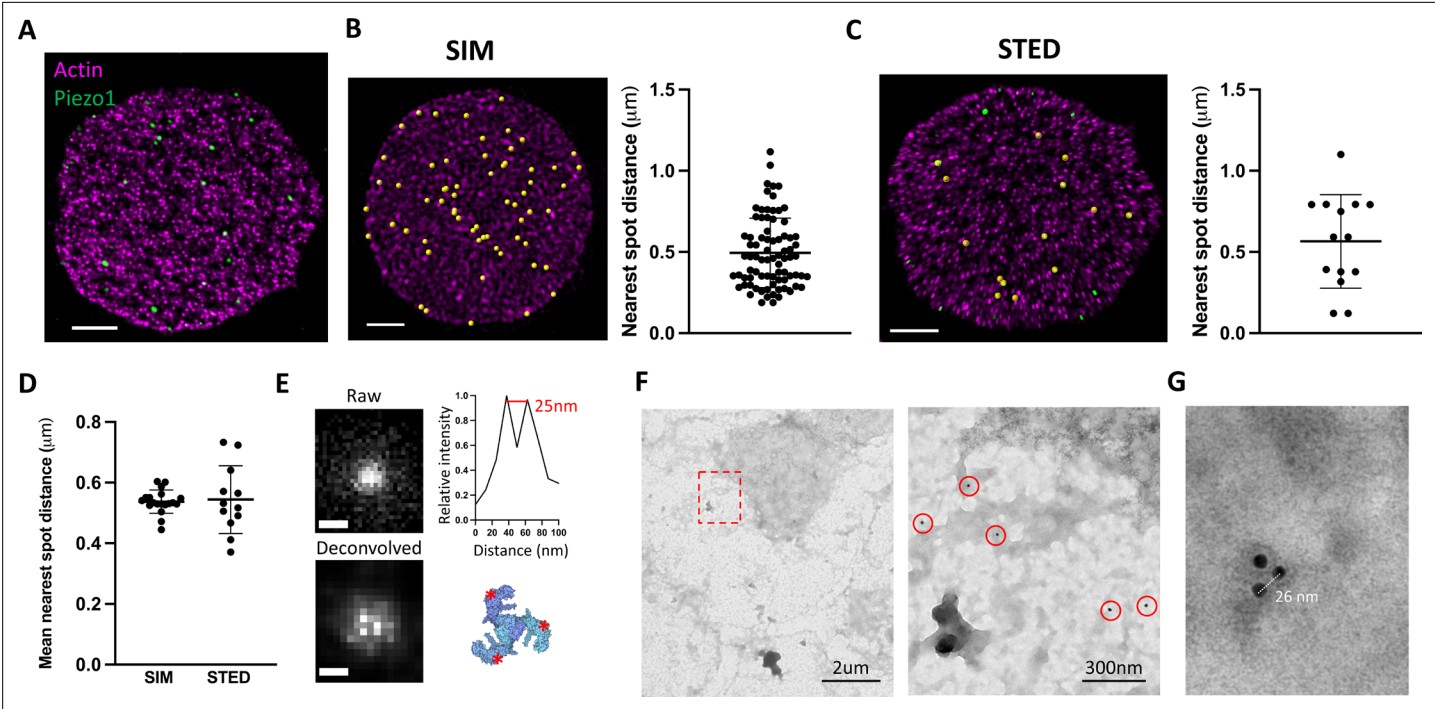

**Figure 2.** Piezo1 does not cluster in red blood cells (RBCs). (**A**) Representative 2D stimulated emission depletion (STED) image of an unroofed RBC immunostained with an anti-HA tag antibody for Piezo1 (green) and labeled with phalloidin STAR 580 for F-actin (magenta). Scale bar, 1 μm. (**B**) 3D reconstruction from SIM imaging of immunostained RBCs, where Piezo1 spots are detected by the spot detector function in Imaris and are colored yellow. The distance from each Piezo1 spot to the nearest Piezo1 spot in this cell was calculated in Imaris and is shown on the right. Scale bar, 1 μm. (**C**) The same analysis as in (B) was carried out on a 2D STED image of an unroofed RBC, but here Piezo1 spots at the edge of the membrane were masked out from spot detection to prevent edge-based artifacts in estimating nearest neighbor distances. Scale bar, 1 μm. (**D**) The mean nearest spot distance was calculated for SIM and STED analyses and plotted. For SIM the mean is 540 ± 37 nm ($n$ = 19 cells) and for STED the mean is 544 ± 106 nm ($n$ = 12 cells). (**E**) (Left) Huygens deconvolution of a Piezo1 puncta, imaged by 2D STED, resolving into a triplet after deconvolution, scale bar is 50 nm. (Bottom right) A top-down view of the Piezo1 channel structure with red asterisks indicating the positions of antibody binding. (Top right) An intensity profile of fluorescent signal from a line-scan of two adjacent bright pixels from the bottom left image, showing a distance of ~25 nm. (**F**) Negative stain electron microscopy of an unroofed RBC immunostained with an anti-HA tag primary antibody and a secondary antibody conjugated to 18-nm gold. Low magnification image of an unroofed RBC (left) with a region highlighted in red imaged at medium magnification (right). 18-nm gold particles corresponding to labeled Piezo1 channels are highlighted by red circles. As in fluorescence microscopy images, Piezo1 channels do not appear to cluster. (**G**) Negative stain electron microscopy image at high magnification of a triple-labeled Piezo1 channel.

The online version of this article includes the following figure supplement(s) for figure 2:

**Figure supplement 1.** Cluster analysis of fluorescent Piezo1 spots in a 2D stimulated emission depletion (STED) image of a labeled unroofed red blood cell (RBC) membrane.

**Figure supplement 2.** 2D STED and Huygens deconvolution analysis of immunolabelled Piezo1 in unroofed RBCs.

*et al., 2010*) shows no statistical difference between the observed spot distribution and a simulation of randomly distributed spots (*Figure 2—figure supplement 1*).

Optimizing 2D STED image acquisition parameters, including imaging over multiple *Z* planes and employing adaptive illumination (*Heine et al., 2017*), as well as image processing through spherical aberration correction and iterative deconvolution (Huygens Professional) (*Schoonderwoert et al., 2013*) enabled an improvement of the lateral FWHM resolution toward ~25 nm (*Figure 2E*), approximating the diameter of the Piezo channel. Post-deconvolution, we observed that a significant number of fluorescent spots resolved into triplets of bright spots, consistent with labeling each arm of the trimeric structure of Piezo1 (*Saotome et al., 2018*; *Guo and MacKinnon, 2017*; *Figure 2E* and *Figure 2—figure supplement 2*). In a single field of view, we can often observe multiple triple-labeled Piezo1 channels (*Figure 2—figure supplement 2*). Line-scan analysis of the intensity between resolved bright pixels after deconvolution yields a pixel size-limited distance of ~25 nm (*Figure 2E*). This distance is consistent across analysis of all bright puncta that resolve into triplets, which we would not expect if these puncta represented multiple channels clustering.

Further, Piezo1 puncta never resolve beyond triplets, consistent with single Piezo1 channels where all three antibody-binding sites are occupied, and not multiple trimeric Piezo1 channels clustered tightly together. The occurrence of triple-labeled Piezo1 in our STED data suggests we are toward the upper limit of labeling. An important corollary to this conclusion is that, at least if the probability of observing a given number of antibodies bound to each Piezo1 channel is non-cooperative, we should expect only a small fraction of Piezo1 channels to be unlabeled, giving us confidence in our estimation of Piezo1 channel number per RBC (*Figure 1C*). This settles inconsistencies in the available quantitative proteomics analyses of the RBC proteome (*Bryk and Wiśniewski, 2017*; *Ravenhill et al., 2019*) and is in line with a study of highly purified erythrocytes, which estimates approximately one hundred Piezo1 channels per cell (*Gautier et al., 2018*). We further analyzed Piezo1 distribution in unroofed red blood cells by negative stain electron microscopy using an 18-nm gold-conjugated secondary antibody (*Figure 2F, G*). Gold particles were distributed over the unroofed RBC membrane without apparent clustering, comparable to the distribution of fluorescent spots in STED images. Gold particles were not observed in RBCs from control mice. By negative stain electron microscopy, we were also able to observe triplets of gold particles (*Figure 2G*) with an inter-particle distance of ~25 nm, consistent with our 2D STED imaging of labeled trimeric Piezo1 channels. Thus, using multiple imaging approaches, we do not observe Piezo1 clustering in the RBC membrane.

## Piezo1 is enriched at the dimple region of RBC membranes

We were curious whether Piezo1 is evenly distributed throughout the RBC membrane or tends to localize to specific regions of the RBC biconcave disk. Because of its intrinsic curvature, it has been hypothesized that Piezo1 concentrates in the curved dimple of the RBC membrane (*Svetina et al., 2019*). Other force-related proteins such as myosin IIA exhibit a non-uniform membrane density and are enriched at the dimple region (*Alimohamadi et al., 2020*). Using 2D slices from 3D-SIM images of intact RBCs we segmented the cells into the dimple and rim regions based on F-actin staining of the membrane (*Figure 3A*) and quantified the number of Piezo1 puncta per $\mu m^2$ surface area of each region using the cell imaging software Imaris (*Figure 3B*). The dimple accounts for about 17% of the total RBC surface area. We find that the whole RBC and rim have similar Piezo1 puncta densities (0.50 ± 0.16 and 0.46 ± 0.17 $\mu m^{-2}$, respectively) while at the dimple region the density is higher (0.80 ± 0.31 $\mu m^{-2}$). Thus, there is approximately a twofold enrichment of Piezo1 at the dimple region of RBC membranes compared to the rim (*Figure 3B*). This enrichment of Piezo1 at the dimple region of the RBC membrane is not due to enzymatic deglycosylation of RBCs, which we used to increase antibody binding to HA-tagged Piezo1 (*Figure 3—figure supplement 1*). To strengthen the hypothesis that it is the intrinsic curvature of Piezo1 that biases it toward the curved dimple region of the RBC we also analyzed the dimple versus rim ratio of two other RBC membrane proteins that are not intrinsically curved: the Gardos channel (*Lee and MacKinnon, 2018*) and Band3 (*Vallese et al., 2022*; *Xia et al., 2022*; *Arakawa et al., 2015*). We find no statistically significant enrichment of the Gardos channel (KCNN4) or Band3 in the dimple of RBCs (*Figure 3C*). Further, we find no statistical enrichment of actin at the dimple, indicating that the actin–spectrin meshwork is not more densely packed there (*Figure 3C*).

Because we observe a range of relative enrichments of Piezo1 at the dimple we asked whether this correlates with the degree of RBC biconcavity, since we also observe a range of biconcavities via F-actin staining of the RBC membrane (*Figure 3—figure supplement 1*). The degree of biconcavity was parameterized as the ratio of the maximum to minimum heights of RBCs at a central *XZ* slice (*Figure 3D*). Indeed, with increasing RBC biconcavity, we observe a greater enrichment of Piezo1 at the dimple (*Figure 3D*). If the distribution of Piezo1 in the RBC membrane is determined by the coupling of the channels' intrinsic curvature to the global curvature of the membrane we would expect an unfavorable distribution of Piezo1 into membrane protrusions where the membrane curvature is opposing to that of Piezo1's. Treatment of RBCs with NaSalicylate, which preferentially binds into the outer membrane leaflet, generates echinocytes (*Li et al., 1999*), RBCs that are characterized by 'thorny' membrane protrusions. 3D-SIM imaging of labeled echinocytes qualitatively shows a depletion of Piezo1 channels from such membrane protrusions relative to regions of negative membrane curvature, further supporting a mechanism of curvature coupling between the intrinsic curvature of Piezo1 and the surrounding membrane (*Figure 3—figure supplement 2*).

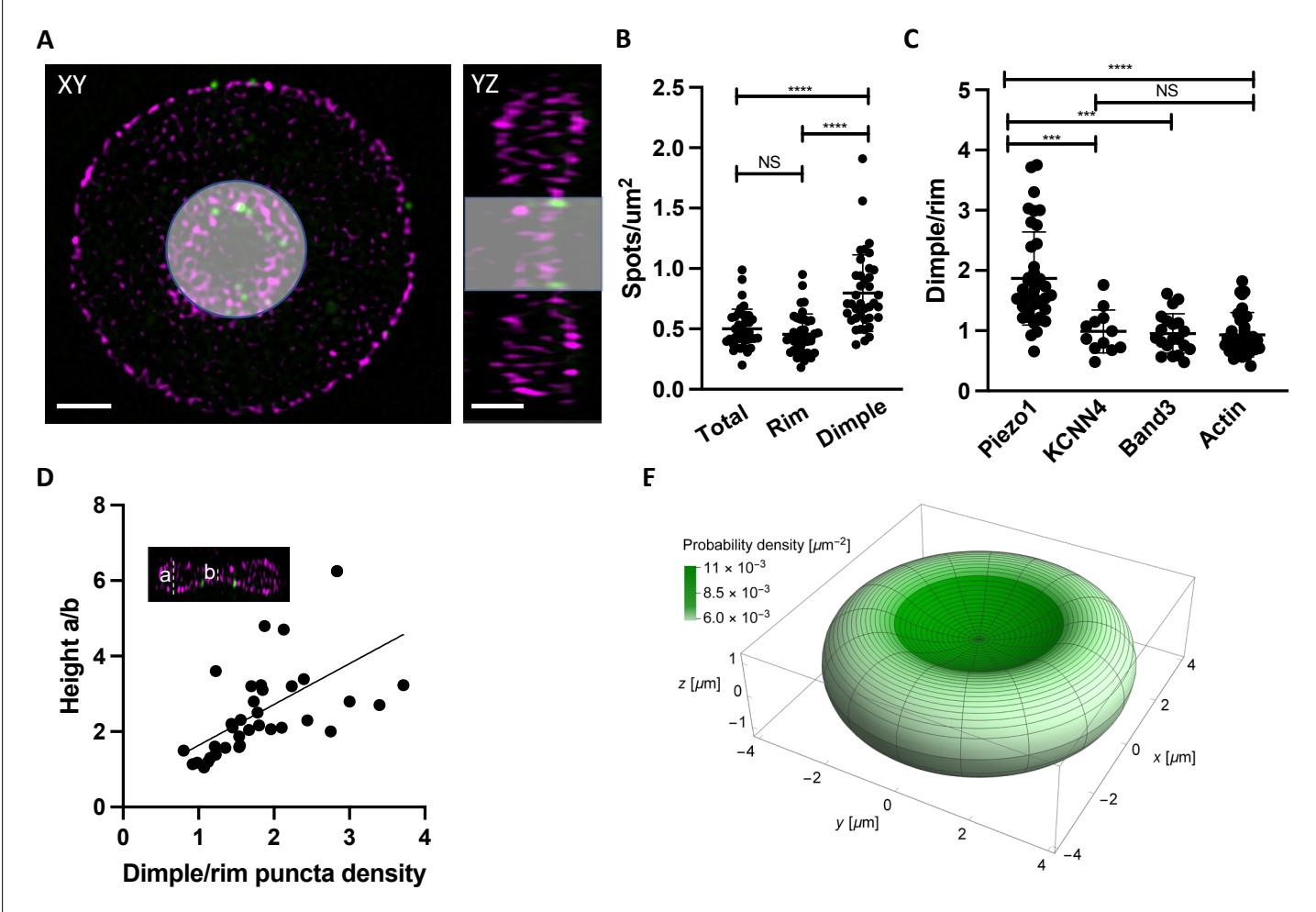

**Figure 3.** Piezo1 localization is enriched in the curved dimple regions of the red blood cell . (**A**) Two dimensional slices in *XY* and *YZ* views from SIM images illustrating volume segmentation of RBCs and Piezo1 puncta distribution after labeling with an antibody against Piezo1-HA (green) and rhodamine–phalloidin for F-actin (magenta). Cells were segmented and masked in Imaris as shown. Scale bar, 1 µm. (**B**) Distribution of Piezo1 puncta in RBCs. The RBC dimple has a higher density of Piezo1 puncta than whole RBCs (Total) (p = <0.0001) and the rim region (p = <0.0001) by the one-way analysis of variance (ANOVA) test with Tukeys post hoc analysis, as indicated on the graph by asterisks where \*\*\* indicates p<0.001 and \*\*\*\* indicates p<0.0001. N = 39 cells (**C**) The ratio of Piezo1 spot density (1.87 ± 0.76, N = 39 cells) in the dimple and rim region of RBCs compared to that of KCNN4 (0.99 ± 0.36, N = 12 cells), Band3 (0.95 ± 0.33, N = 20 cells), and Actin (0.93 ± 0.36, N = 31 cells). The dimple over rim ratio is higher for Piezo1 than KCNN4 (p = 0.0003) and Band3 (p = 0.0003) and Actin (p < 0.0001). (**D**) Scatterplot of ratio of Piezo1 puncta density in dimple over rim plotted against RBC biconcavity, measured as height ratio a/b (see inset). N = 37 cells. Data are fit by linear regression with equation y = 1.08x + 0.55, $R^2$ = 0.34 and p = 0.0001. (**E**) Probability per unit area for finding a Piezo1 channel at a given RBC membrane location, calculated from a physical model of Piezo1 curvature coupling (see methods) and plotted over the RBC membrane surface. The RBC shape corresponds to Beck's model of for RBC discocytes (*Beck, 1978*).

The online version of this article includes the following figure supplement(s) for figure 3:

**Figure supplement 1.** Examples of two cells with different biconcavities and analysis of PNGase F treatment on Piezo1 distribution.

**Figure supplement 2.** Piezo1 is depleted from membrane regions of high positive curvature in echinocytes.

The relative abundance of Piezo1 in the dimple and positive correlation with the degree of biconcavity indicates that RBC membrane curvature might be the primary determinant of the Piezo1 distribution. The intrinsic curvature of Piezo1 suggests a possible mechanism, based on 'curvature coupling', for bias toward the dimple. To examine the plausibility of this mechanism, we developed a simple physical model of Piezo1 curvature coupling in which Piezo1 is represented as a spherical cap with area 450 nm$^2$ and approximate radius of curvature 42 nm, the estimated asymptotic value in a planar membrane (*Haselwandter et al., 2022a*). To describe the RBC shape, we used the model of *Beck, 1978* and calculated the mean curvature at each point on the surface (*Methods*). Then, using

the Helfrich functional (*Helfrich, 1973*), we calculated the minimum shape energy of the membrane surrounding Piezo1 in a vesicle that, if spherical, would have a radius of curvature equal to the reciprocal of the mean curvature at that point on the RBC surface. This procedure associates each point on the RBC surface with a membrane curvature energy which, when applied to the Boltzmann distribution equation, yields a probability distribution for the density of Piezo1 on the membrane surface. Further details of this calculation are outlined in *Methods*. The surface distribution of Piezo1 calculated according to this method is shown in *Figure 3E*. The curvature coupling model predicts an approximately two-fold increase in the density of Piezo1 in the dimple relative to the rim, close to the experimentally observed ratio of densities. In the Beck model of RBC shape, the mean curvature at any point on the RBC surface ranges from 0.24 µm$^{-1}$ (favorable to Piezo1 in the dimple) to about −0.52 µm$^{-1}$ (unfavorable to Piezo1 in the rim). Because these mean curvatures for the RBC membrane are very small (compare these values to the approximate intrinsic curvature of Piezo1, 1/42 nm ≈24 µm$^{-1}$), it was not immediately clear to us whether Piezo1 would be sensitive to surface curvature in the RBC. The correspondence between experiment and theory suggests that Piezo1 is indeed sensitive to RBC shape and adopts an equilibrium distribution by a mechanism of curvature coupling. If one thinks about this result, the sensitivity of Piezo1 distribution to the global curvature of the RBC membrane implies

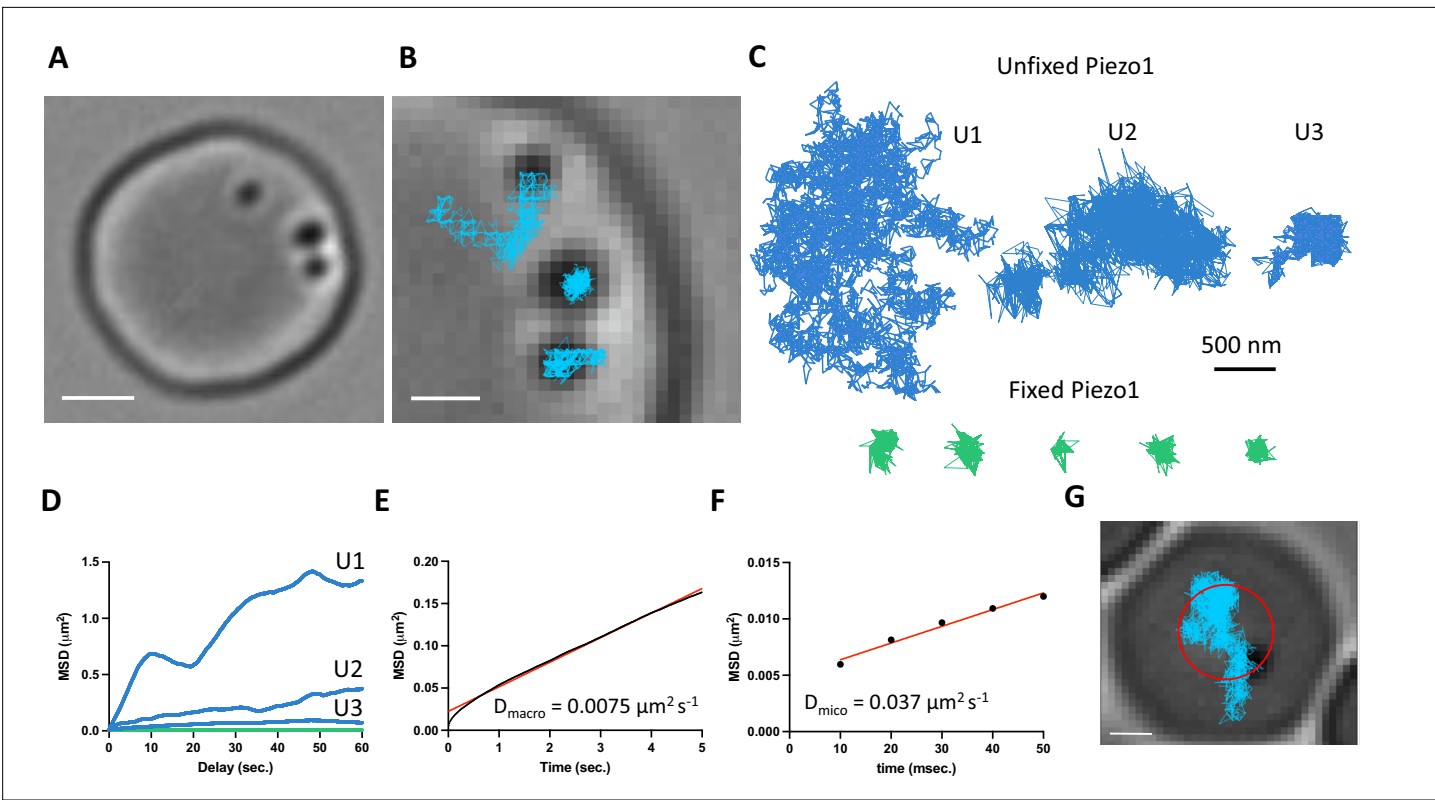

**Figure 4.** Gold-particle tracking of Piezo1 reveals its lateral mobility in the red blood cell membrane. (**A**) Example of a RBC imaged under DIC microscopy showing three 40-nm gold particle-labeled Piezo1 channels. Scale bar, 2 µm. (**B**) Close-up image of the gold particles with trajectories from a 2-min recording at 100 Hz frequency. Scale bar, 1 µm. (**C**) Representative trajectories corresponding to distinct Piezo1 diffusive behaviors in unfixed conditions, shown in blue (trajectories U1, U2, and U3). The green trajectories correspond to labeled Piezo1 in cells fixed with paraformaldehyde (PFA). (**D**) MSD against time for the first 60 s of the trajectories U1, U2, and U3 in C corresponding to unfixed conditions (blue) and an average of n = 5 fixed trajectories (green). (**E**) MSD against time for the first 5 s of an average of 14 trajectories. The data are fit to a straight line with a slope corresponding to a 2D diffusion coefficient of 0.0075 µm$^2$ s$^{-1}$. $R^2$ for linear fit to data is 0.99. (**F**) MSD against time for the first 50 ms of an average of 14 trajectories. The data are fit to a straight line with a slope corresponding to a 2D diffusion coefficient of 0.037 µm$^2$ s$^{-1}$. $R^2$ for linear fit to data is 0.98. (**G**) Image of a RBC with a single gold-labeled Piezo1 channel that was recorded for 1 hr at 1 Hz frequency. Superimposed are the final tracking result (blue) and a circle (red) indicating the approximate segmentation of the RBC dimple. Scale bar, 1 µm.

The online version of this article includes the following figure supplement(s) for figure 4:

**Figure supplement 1.** The observed diffusion behavior of Piezo1 is consistent with the organization of the actin–spectrin skeleton in red blood cells.

**Figure supplement 2.** The diffusion behavior of Piezo1 is comparable in the dimple and rim of the red blood cell.

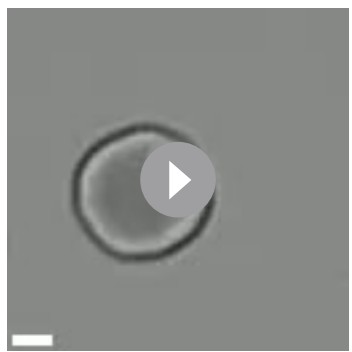

**Video 1.** Example video recording of a RBC from a *Piezo1*^HA mouse immunolabeled with three 40-nm gold particles and imaged under DIC. The recording was made at 100 Hz frequency and lasted 2 min. Scale bar, 1 µm. Video is sped up 2x.

https://elifesciences.org/articles/82621/figures#video1

**Video 2.** Example video recording of a RBCfrom a *Piezo1*^HA mouse immunolabeled with a 40-nm gold particle and imaged under differential interference contrast (DIC). The recording was made at 1 Hz frequency and lasted 1 hr. Scale bar, 1 µm. Video is sped up 100x.

https://elifesciences.org/articles/82621/figures#video2

that the membrane must be relatively smooth. Indeed, this implication is consistent with electron microscopic images of healthy RBCs (*Marikovsky and Danon, 1969*; *Lewis et al., 1968*).

## Piezo1 is mobile within the red blood cell plasma membrane

The above distribution of Piezo1 according to the curvature coupling model suggests that Piezo1 ought to be laterally mobile so that it can explore the curvature landscape of the RBC membrane. However, the quasihexagonal actin–spectrin meshwork and large junctional protein complexes that make up the RBC membrane layer are known to corral and confine membrane protein diffusion (*Sheetz, 1983*). Additionally, at least some membrane proteins, including roughly one third of Band3, are essentially immobilized by direct binding to actin–spectrin (*Tomishige et al., 1998*; *Tsuji et al., 1988*). We studied Piezo1 mobility in the RBC membrane with SPT using 40-nm gold nanoparticles conjugated to a secondary antibody under differential interference contrast (DIC) microscopy (*Wu et al., 2019*). The fast time resolution and ability to record over minutes without photobleaching has advantages for studying confined membrane protein diffusion (*Tomishige et al., 1998*; *Daumas et al., 2003*). Gold particles could be unambiguously identified on RBC membranes (*Figure 4A* and *Video 1*). We imaged at 100 Hz for 2 min. In recordings lasting 2 min we observed a range of Piezo1 diffusion trajectories (*Figure 4B–D*). Most trajectories exhibited confined diffusion, although sometimes Piezo1 appeared to be freely diffusing. Occasionally, we observed both types of diffusion in a single recording. In all cases, Piezo1 was more mobile than in control cells chemically fixed with paraformaldehyde, when analyzed by a mean squared displacement (MSD) analysis (*Figure 4D*). For confined trajectories, we find that the confinement diameter is in the range of 200–600 nm. Although limited by the spatial resolution of our recordings, this range is largely consistent with the organization of the actin–spectrin meshwork we observe by 2D STED imaging of unroofed RBCs stained for actin or spectrin (*Figure 4—figure supplement 1*). In addition to smaller corrals of intact actin–spectrin, we find voids deficient in actin or spectrin staining, approximately 200–500 nm in size (*Figure 4—figure supplement 1*). Such voids in the actin–spectrin skeleton have also been observed by Stochastic Optical Reconstruction Microscopy (STORM) studies on RBCs (*Pan et al., 2018*). Plotting the average MSD of the first 5 s of Piezo1 tracks yielded an apparent macroscopic diffusion coefficient of 0.0075 $\mu m^2 \ s^{-1}$ (*Figure 4E*), which is similar to that of the mobile population of Band3 (*Tomishige et al., 1998*) and likely reflects occasional localized periods of confinement within actin–spectrin corrals over time. Microscopic diffusion was also analyzed by plotting the average MSD of the first 50 ms of Piezo1 tracks before the channel will tend to become localized within confinement zones. This analysis yielded an apparent diffusion coefficient of 0.037 $\mu m^2 \ s^{-1}$, consistent with measurements of freely diffusing Piezo1 in mouse neural stem progenitor cells (*Ellefsen et al., 2019*). The diffusion coefficient of a free ~50-nm gold nanoparticle in solution has been reported as ~1 $\mu m^2 \ s^{-1}$ (*Giorgi et al., 2019*), two orders of magnitude faster than the estimated microscopic diffusion coefficient of gold-labeled Piezo1 in our experiments. It is possible that a 40-nm gold particle may somewhat slow diffusion of membrane-embedded Piezo1, leading to an underestimation of its diffusion coefficient by our calculations. Nevertheless, Piezo1 appears laterally mobile within the RBC membrane.

Based on our imaging experiments of fixed RBCs that show an enrichment of Piezo1 in the dimple (*Figure 3*), we tracked Piezo1's preference for the dimple versus rim in real time by recording for an hour at 1-Hz frequency (*Figure 4G* and *Video 2*). We approximated the dimple as a circle with radius 1.13 μm, comparable to our segmentation based on F-actin staining (*Figure 3*) and calculated the frequency at which gold-labeled Piezo1 is within 1.13 μm of an origin defined as the center of the RBC (*Figure 4G*). We find that Piezo1 is observed within the dimple 81% of the time during this 1 hr recording, consistent with a preference of Piezo1 for the dimple over the rim. This observed preference for the RBC dimple was not due to slower diffusion here: analysis of tracks that could easily be segmented as at the dimple or the rim yielded macroscopic diffusion coefficients of 0.0086 and 0.0074 μm² s⁻¹, respectively, which were not statistically different (*Figure 4—figure supplement 2*). If Piezo1 were to bind to specific sites in the dimple we would expect the diffusion coefficient to be smaller there. As the diffusion coefficient is not smaller, we can understand the observed bias toward the dimple as resulting from curvature coupling between Piezo1's shape and the membrane's curvature.

## Piezo1 is not bound to the actin–spectrin meshwork

To further explore the behavior of Piezo1 in RBC membranes we analyzed its spatial proximity to the actin–spectrin meshwork by fluorescence microscopy. Proteins including ankyrin and protein 4.1 bind both to components of the meshwork and to proteins in the membrane, most commonly Band3 proteins, organizing the RBC membrane into functional microdomains composed of channels and transporters tethered to the actin–spectrin skeleton (*Lux, 2016*). Using co-immunostaining of unroofed RBC membranes, we asked whether Piezo1 is associated with these well-known complexes. Because of the small size of RBCs and the density of actin–spectrin at the membrane, super-resolution microscopy was necessary to address this question. Indeed, others have claimed that Piezo1 does colocalize with spectrin in red blood cells (*Dumitru et al., 2021*), but this study used diffraction-limited confocal microscopy, whereby every red blood cell membrane protein would be expected to show some degree of colocalization with spectrin. We employed 2D STED to image unroofed RBCs. To analyze spatial proximity of two fluorescent signals in our STED datasets we utilized optimal transport colocalization (OTC) (*Tameling et al., 2021*) analysis, which is more appropriate for STED

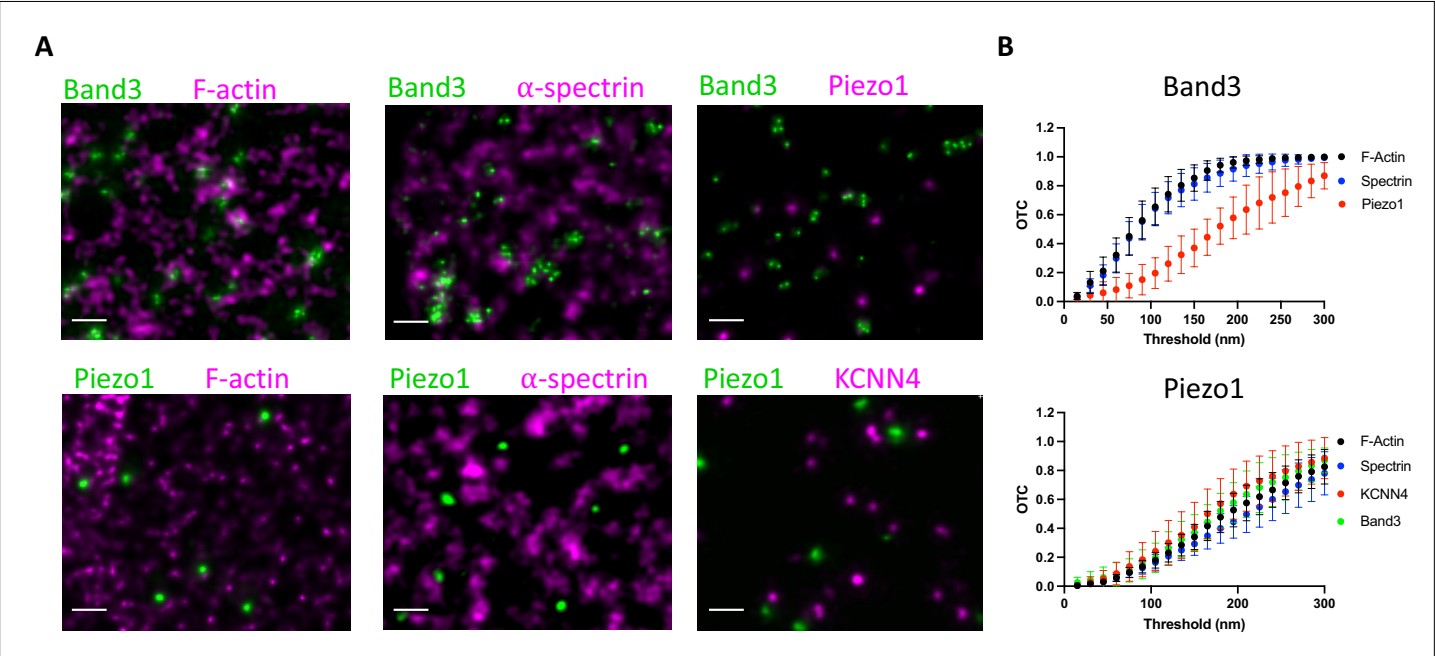

**Figure 5.** Piezo1 does not appear to interact with the membrane skeleton. (**A**) Exemplary STED microscopy images assessing interactions of Band3 (upper panels) and Piezo1 (lower panels) with known members of the RBC membrane skeleton as well as with each other and with KCNN4. Close spatial proximity for Band3 with actin and spectrin can be observed in STED images but not for Piezo1. Scale bar, 200 nm. (**B**) OTCanalysis including 95% confidence error bars of STED microscopy image sections (Band3/F-actin, *n* = 11; Band3/Spectrin, *n* = 10; Band3/Piezo1, *n* = 7; Piezo1/F-actin, *n* = 10; Piezo1/Spectrin, *n* = 7; Piezo1/KCNN4, *n* = 8).

microscopy than traditional pixel-based methods. First, we validated our experimental approach by co-immunostaining proteins of known complexes at the RBC membrane. There are three distinct known populations of Band3: one linked to spectrin via ankyrin, another bound to actin via protein 4.1 and other bridging proteins, and finally proteins that remain untethered to the actin–spectrin meshwork (*Lux, 2016*; *Kodippili et al., 2009*). A significant fraction of Band3 fluorescent puncta are observed as overlapping or directly adjacent to both F-actin and spectrin puncta by STED microscopy, indicating close spatial proximity (*Figure 5A*). In contrast, we do not observe a significant spatial proximity of Band3 and Piezo1, suggesting that if Piezo1 is tethered to actin–spectrin it is not as part of the Band3 complex. This is quantified by OTC analysis, where we plot the fraction of pixel intensities that is matched (colocalized) at distances less than the defined threshold distance. Following Nyquist sampling, the pixel size of 12.5 nm served as the lower limit of this threshold and the upper limit was set to 300 nm (*Figure 5B*). At a threshold of 100 nm, for example, 0.6 or 60% of signals from Band3 complexes are considered proximal enough to specify as colocalized with both actin and spectrin, consistent with the two aforementioned well-known populations of Band3 complexes. By contrast, only 0.15 or 15% of Piezo1 signals are considered proximal enough to specify as colocalized with Band3. We also co-immunostained Piezo1 with F-actin or spectrin to ask whether it is tethered to the RBC membrane skeleton. We do not observe significant spatial proximity between Piezo1 to either F-actin or spectrin by STED microscopy (*Figure 5A*). At a threshold of 100 nm we observe an OTC score of only 0.1 for Piezo (*Figure 5B*). Thus, our data show that unlike Band3, Piezo does not tend to exhibit close spatial proximity to actin or spectrin and is unlikely to bind to the membrane skeleton, consistent with its observed mobility in tracking experiments.

Based on a proposed mechanism of Piezo1-mediated $Ca^{2+}$ influx activating the Gardos channel (*Cahalan et al., 2015*) (KCNN4), we also analyzed spatial proximity of these two proteins by STED. Piezo1 and KCNN4 do not appear to form complexes in RBCs (*Figure 5A, B*), but it should be noted, given the small size of the RBC, that colocalization of Piezo1 and KCNN4 is not necessary for a functional coupling of the ion fluxes associated with these two channels (see *Discussion*).

## Discussion

Since its discovery (*Coste et al., 2010*), Piezo1 has been identified as a primary mechanosensitive molecule in a wide range of force-sensing physiological processes from cell and organ development (*Sun et al., 2019*; *Li et al., 2014*; *Ranade et al., 2014*) to blood pressure regulation (*Zeng et al., 2018*) and stem cell lineage choice (*Pathak et al., 2014*). How Piezo1 is organized in different cell types and how this might pertain to the sensing of different types of physiological force is currently not well understood. Here we studied Piezo1 organization in RBCs, both because of the clear importance of Piezo1 in RBC physiology (*Bae et al., 2013*; *Zarychanski et al., 2012*; *Cahalan et al., 2015*) and because of the unique cellular structure of RBCs, particularly their biconcave disk shape.

At the smallest level of Piezo1 organization, that is with itself, we find that Piezo1 does not cluster in the RBC membrane. By 3D-SIM microscopy (*Figure 1*) we observe diffraction-limited fluorescent spots that could technically represent multiple, closely located, Piezo1 channels. To distinguish these spots as individual channels, we needed to employ 2D STED microscopy (*Figure 2*). As the power of the depletion laser was increased to improve resolution, the size of each spot decreased correspondingly, but never resolved into multiple spots, even when the achieved resolution was near the size of the Piezo1 channel. Further, after Huygens deconvolution post-processing of our 2D STED data we could resolve triple-antibody-labeled Piezo1 channels, with an inter-subunit diameter consistent with the channel size. Thus, we conclude that each fluorescent spot, whether viewed by 3D-SIM or 2D STED, corresponds to a single Piezo1 channel. The average distance between Piezo1 channels is approximately 550 nm, although occasionally we observe two to three channels within 100-nm distances. This is in good agreement with studies of neuro2A cells, which natively express Piezo1, that also found that Piezo1 channels do not tend to cluster (*Lewis and Grandl, 2021*). While clustering could affect Piezo1 gating through Piezo's membrane footprint, the precise coupling of Piezo1 gating to clustering would depend on the specific scenario considered, and clustering does not necessarily facilitate gating (*Haselwandter and MacKinnon, 2018*). A relatively homogenous membrane distribution of Piezo1 might be well suited to RBCs that are tumbling through solution, experiencing shear forces from all angles, and may contribute to the remarkable plasticity of RBC shapes under stress

(*Dupire et al., 2012*). Of course, our data on RBCs do not rule out the possibility of Piezo1 clustering in other cell types.

Because of the intrinsic curvature of Piezo1 we were curious about its distribution as a function of the curvature of the RBC membrane. Experimentally we were able to ask this question through 3D-SIM microscopy, which yielded sufficiently high-resolution spatial information on Piezo1 channels while enabling easy assignment of the RBC dimple and rim membrane regions (*Figure 3*). Per $\mu m^2$ area of membrane we find an almost twofold enrichment of Piezo1 in the dimple relative to the rim. Our experimental data can be rationalized by a physical model of curvature coupling between the intrinsic curvature of the Piezo1 dome and the surrounding membrane curvature of the RBC along its biconcave surface (see *Methods*). Whilst the curved biconcave shape of RBCs is unique, the good agreement between our experimental data and theoretical prediction based on energetic curvature coupling would suggest an inherent relationship between Piezo's intrinsic curvature and the surrounding membrane that is broadly relevant to other cell types. This is because the curvature coupling theory, which was developed with experiments on lipid bilayer vesicles, depends only on the bending elasticity of membranes and the intrinsic curvature of Piezo1, not on the unique shape of RBCs. It will be interesting to explore whether Piezo1, for example, is enriched in membrane invaginations or other curvature-matched membrane regions in different cell types. Recently, a study analyzing overexpressed Piezo1 in HeLa cells showed depletion of the channel from membrane filopodia, consistent with a mechanism of curvature coupling whereby the channel unfavorably distributes into membrane regions of high opposing curvature (*Yang et al., 2022*).

Curvature sensing of membrane proteins by curvature matching has been well documented for other intrinsically curved proteins such as Bin/Amphiphysin/Rvs (BAR) (*Simunovic et al., 2015*; *Tsai et al., 2021*) proteins. More recently, it has been shown that glycosylation can determine the membrane curvature sensing properties of some proteins (*Lu et al., 2022*). It is unclear from our current data whether the enrichment of Piezo1 in the RBC dimple is simply a corollary of the curved structure of Piezo1, which determines the channel's tension sensitivity as described by the membrane dome model (*Guo and MacKinnon, 2017*; *Haselwandter and MacKinnon, 2018*), or whether it also has some important consequences for physiological force sensation of RBCs. In this context, it is noteworthy that the force-generating protein myosin IIA also shows an enrichment in the RBC dimple relative to the RBC rim (*Alimohamadi et al., 2020*).

Consistent with its ability to equilibrate along the RBC surface as a function of membrane curvature, we find that Piezo1 exhibits lateral mobility within the RBC plasma membrane. We used gold nanoparticle tracking under DIC (*Figure 4*) so we could record for minutes to an hour without being limited by photobleaching and at fast time resolution (100 Hz). This enabled us to record both the confined (<500 nm) diffusion of Piezo1 as well as Piezo1's 'hopping' into more freely diffusing behavior. The spatial scales associated with the confined diffusion regime most likely reflect the compartment sizes of the actin–spectrin meshwork we observe by 2D STED. Such a model of confined diffusion set by actin–spectrin is well documented in RBCs (*Tsuji et al., 1988*) and other cell types (*Fujiwara et al., 2016*). Similar to observations by others using STORM microscopy (*Pan et al., 2018*), we observe ~200–500 nm voids that are deficient in actin or spectrin signal by 2D STED. Similar nanoscale 'defects' in the actin–spectrin meshwork have also been observed by atomic force microscopy imaging of RBCs (*Nowakowski et al., 2001*) and may serve as weak points in the meshwork that facilitate fast RBC remodeling during shear stress. Such defects may also permit faster diffusion of transmembrane proteins such as Piezo1.

Our double labeling experiments suggest that Piezo1 does not form a complex with actin or spectrin, which is consistent with Piezo1's confined diffusion within the actin–spectrin meshwork compartments, and should be contrasted with the near-immobilization observed for proteins such as CFTR and Band3 (*Kodippili et al., 2009*, 3; *Haggie et al., 2006*), which both form complexes with actin. Additionally, our data are not consistent with Piezo1 being tethered to the underlying actin and activated by a 'force-from-filament' mechanism as has been traditionally described (*Cox et al., 2019*; *Zhang et al., 2015*). Of course, the actin–spectrin meshwork organizes the RBC membrane into compartments whose boundary conditions could impact how changes in local curvature and lateral tension of the membrane gate Piezo1. This could also explain how chemical agents that disrupt the actin cytoskeleton affect Piezo1 currents in other cell types (*Gottlieb et al., 2012*; *Wang et al., 2022*), that is through changes in organization of the membrane rather than disruption of a direct tether to the channel itself.

Additionally, we asked whether Piezo1 might be colocalized with the $Ca^{2+}$-activated $K^+$ (Gardos) channel. $Ca^{2+}$ flux into the RBC through Piezo1 and subsequent activation of the Gardos channel is thought to be the initial step in mechanically activated RBC volume regulation (*Danielczok et al., 2017*; *Cahalan et al., 2015*). We do not, however, observe significant spatial proximity between Piezo1 and Gardos channels. Given the small size of RBCs, approximately 1 µm thick in the dimple and 2–3 µm thick in the rim, and the Piezo1 density of around 0.5 per $µm^2$, it seems that colocalization is probably not necessary for rapid opening of the Gardos channel following Piezo activation. For diffusion in three dimensions, the mean dispersion time can be approximated by $\tau = \frac{x^2}{6D}$, $x$ the mean distance and $D$ the diffusion coefficient. The apparent diffusion coefficient for intracellular $Ca^{2+}$ is in the range 13–65 $µm^2/s$ (*Nakatani et al., 2002*), meaning that $Ca^{2+}$ should equilibrate within tens of ms within a RBC. In larger cell types, colocalization of Piezo1 with other channels might be necessary to elicit a functional coupling of ion fluxes. In rat ventricular myocytes, for example, Piezo1 has recently been shown, by diffraction-limited confocal microscopy, to colocalize with transient receptor melastatin (TRPM4) channels (*Yu et al., 2022*).

The experimentally determined values of the intrinsic curvature and the bending stiffness of Piezo1 in lipid bilayers enable us to estimate changes in Piezo1 curvature at the dimple versus the rim of a RBC membrane (*Haselwandter et al., 2022a*). We estimate a radius of curvature of 41.4 nm at the dimple and 42.7 nm at the rim and thus the curvature differences across the unperturbed RBC membrane are, alone, insufficient to substantially change the shape of Piezo1, and thus are not expected to open its pore. However, that Piezo1 distributes on the surface of RBCs by sensing the membrane curvature, by corollary, means that it must have the capacity to sense lateral membrane tension by a force-through-membrane mechanism. To see this, consider that Piezo1 has been shown to flex, that is, change its curvature, as a function of local membrane curvature (*Haselwandter et al., 2022b*). Consider also that membrane tension changes membrane curvature by flattening it. In this manner, curvature sensing and lateral membrane tension sensing are inextricably linked. This linkage lies at the heart of the membrane dome model of Piezo gating (*Guo and MacKinnon, 2017*; *Haselwandter and MacKinnon, 2018*; *Haselwandter et al., 2022b*; *Haselwandter et al., 2022a*). When RBCs squeeze through small capillaries, the membrane shape as well as the transmembrane pressure change transiently. The resulting changes in membrane curvature and lateral membrane tension can both, according to the membrane dome model of Piezo gating, open Piezo1's pore to trigger $Ca^{2+}$ entry.

## Methods

### HA-tag knock in mice

C57BL/6 background mice heterozygous for an HA tag knocked into the coding sequence after amino acid position 893 of *Piezo1*, generated by CRISPR/Cas9 technology, were purchased from Jackson Laboratories. Heterozygous mice were bred to generate mice homozygous for the HA tag knock in and were genotyped by PCR amplification around the 893 insertion site and gel electrophoresis analysis of the amplified band size.

### Preparation of blood

Adult mice (>3 months old) were anesthetized by isoflurane inhalation and retro-orbital bleeds were taken using a heparinized micro-hematrocrit capillary tube (Fisherbrand). Whole blood was washed in Dulbecco's phosphate-buffered saline (DPBS) three times by centrifugation at 1000 × *g* to separate RBCs from the buffy coat.

### Electrophysiological recordings

Washed RBCs were diluted 1:1000 and plated on uncoated Petri dishes containing bath solution consisting of 140 mM NaMethanesulfonate, 10 HEPES pH 7.4. RBCs were imaged with a Nikon eclipse DIC microscope at ×40 magnification. Pipettes of borosilicate glass (Sutter Instruments; BF150-86-10) were pulled to ~15–20 MΩ resistance with a micropipette puller (Sutter Instruments; P-97) and polished with a microforge (Narishige; MF-83). The pipette was filled with identical bath solution. Recordings were obtained with an Axopatch 200B amplifier (Molecular Devices), filtered at 1 kHz and digitized at 10 kHz (Digidata 1440A; Molecular devices). Gigaseals were obtained by applying a suction pulse

(−10 mm Hg) using a high-speed pressure clamp (ALA scientific). Holding at negative voltages often facilitated seal formation. To obtain excised inside-out patches, attached cells were lifted briefly to the air–water interface which about 50% of the time would remove the cell but leave an intact inside-out membrane patch in the pipette.

## Immunofluorescence of intact RBCs using 3D-SIM

Washed RBCs were deglycosylated by incubation with 10% PNGase F (New England Biolabs) at 37°C for 2 hr, which we found increased antibody binding. RBCs were then diluted 1:50 in DPBS with 4% PFA and incubated overnight at room temperature to fix. Fixed cells were washed three times in DPBS by centrifugation at $1000 \times g$ for 5 min and then permeabilized in DPBS + 0.3% Triton X-100 for 15 min. Permeabilized cells were then blocked in DPBS + 4% bovine serum albumin (BSA) + 1% normal goat serum (blocking buffer) overnight at 4°C. Permeabilized and blocked RBCs were then incubated with a rabbit anti-HA primary antibody (3724; Cell Signaling Technology) diluted in blocking buffer for 1 hr at room temperature, washed three times in blocking buffer and then incubated in 1:500 dilution of Alexa-488-conjugated goat anti-rabbit secondary (ab150077; Abcam) mixed with ×1 rhodamine–phalloidin (Invitrogen) for 1 hr, followed by washing three times in blocking buffer. For 3D-SIM imaging of KCNN4 and Band3 in RBCs, 1:100 dilution of rabbit anti-Band3 primary (Proteintech) and 1:500 dilution of rabbit anti-KCN44 (76647, Invitrogen) were used. 1:500 dilution of Alexa-488-conjugated goat anti-rabbit (ab150077; Abcam) was used as the secondary along with 1× rhodamine–phalloidin (Invitrogen).

3D-SIM images were acquired using a DeltaVision OMX V4/Blaze system (Cytiva) fitted with an Olympus ×100/1.40 NA UPLSAPO oil objective and Photometrics Evolve EMCCD cameras. 488 and 568 nm laser lines were used for excitation and the corresponding emission filters sets were 528/48 and 609/37 nm, respectively. Image stacks were acquired with an optical section spacing of 125 nm. SI reconstruction and Image Registration were performed with softWoRx v 6.1 software using Optical Transfer Functions (OTFs) generated from Point Spread Functions (PSFs) acquired from 100 nm green and red FluoSpheres and alignment parameters refined using 100 nm TetraSpeck beads (Invitrogen).

Imaris version 9.9 was used for analyzing 3D-SIM images. Detection of Piezo1 fluorescent spots was performed with Imaris spot detection with an estimated $XY$ diameter of 120 nm, based on measurements of fluorescent spots in 2D slices and the approximate $XY$ resolution limit of 3D-SIM experiments. Spots were mostly uniform in size with a maximum $XY$ diameter of 180 nm. The intensity threshold for spot detection was set to between 3000 and 5000 based on excluding detection of weak background fluorescent signal on the coverslip surface.

## Echinocyte formation

Washed RBCs that were degclyosylated were incubated in DPBS + 10 mM NaSalicylate for 30 min at 37°C to generate echinocytes before fixation and immunostaining as described above.

## Immunofluorescence of unroofed RBCs using STED microscopy

Washed RBCs that were deglycosylated as above were plated onto poly-D-lysine coated high-performance coverslips and allowed to adhere at room temperature for 15 min. Plated RBCs were unroofed by applying a stream of 10 ml DPBS through a 20 gauge needle at a ~20° angle (*Swihart et al., 2001*). Unroofed RBCs were fixed in DPBS with 4% PFA for 15 min and then washed three times in DPBS before blocking in blocking buffer for 30 min. This was followed by immunostaining with 1:500 dilution of rabbit anti-HA primary (3724; Cell Signaling Technology) for 1 hr at room temperature, washing three times in blocking buffer and then incubating in 1:500 dilution of anti-rabbit STAR RED secondary (Abberior) mixed with 1× phalloidin STAR 580 (Abberior) for 1 hr. Immunostained unroofed cells were then, again, washed three times in blocking buffer and mounted in uncured Prolong Diamond Antifade mountant (Invitrogen) before imaging. For co-immunostaining experiments, additional antibodies used were as follows: 1:100 dilution of rabbit anti-Band3 primary (Proteintech), 1:100 dilution of mouse anti-alpha 1 Spectrin (ab11751), and 1:500 dilution of rabbit anti-KCNN4 (76647, Invitrogen). For co-immunostaining experiments with rabbit primary antibodies, rat anti-HA (Roche) primary was used to label Piezo1 and in all cases the species-appropriate STAR RED and STAR 580 (Abberior) secondaries were used.

STED microscopy was performed using a Facility Line STED microscope (Abberior Instruments) equipped with Olympus IX83 stand, Olympus UPLXAPO ×100/1.45 NA oil objective, pulsed excitation lasers with time gating (405, 488, 561, and 640 nm) and a 775-nm pulsed STED depletion laser, 4 Avalance Photo Diode detectors, adaptive illumination packages (DyMIN/ RESCue), and a deformable mirror for correction of spherical aberrations. Abberior Imspector software version 16.3.14287-w2129 with Lightbox interface was used for image acquisition. Fluorophore excitation was facilitated at 580 nm (Phalloidin STAR 580) and 640 nm (STAR RED), whereas depletion was achieved using a wavelength of 775 nm. A pixel size of 12.5 nm was used in 2D STED mode. Excitation laser power, depletion laser power, line averaging/accumulation, and pixel dwell time were optimized for balancing signal-to-noise ratio (SNR), and resolution, while minimizing photobleaching. Adaptive illumination techniques RESCue (*Staudt et al., 2011*) and DyMIN (*Heine et al., 2017*) were employed to reduce photobleaching and increase SNR.

Acquired STED images were saved as OBF files, which were deconvolved using theoretical PSF modeled from microscopy parameters in Huygens Professional version 22.04 (Scientific Volume Imaging). Imported images were cropped as needed, microscopic parameters were edited for refractive index mismatch correction, the excitation fill factor was set to 1.38 per software developer recommendations, and the STED saturation factor was tested within a range of 40–80, as per vendor recommendations, for optimal resolution. The deconvolution wizard was used to estimate background manually in the raw images and the acuity option was set for optimal image sharpness. Deconvolved images were exported as 16-bit TIFF files.

Cluster analysis on 2D STED images was performed using the spatial statistics 2D/3D ImageJ plugin (*Andrey et al., 2010*).

## Image analysis in Imaris

Imaris version 9.9 was used for analyzing fluorescent images. Detection of Piezo1 fluorescent spots was performed with Imaris spot detection with an estimated *XY* diameter of 120 nm, based on measurement of fluorescent spots in 2D slices and the approximate *XY* resolution limit of 3D-SIM experiments.

## Negative stain electron microscopy of unroofed RBCs

Washed and deglycosylated RBCs were diluted 1:100 and added to carbon coated 400 mesh copper grids (CF400-CU, Electron Microscopy Sciences) that had been glow discharged and pre-coated in poly-D-lysine solution. After 30 min incubation, RBCs were unroofed in DPBS as described above. Unroofed RBCs were blocked for 30 min at room temperature in blocking buffer and then incubated with 1:200 rabbit anti-HA primary antibody (3724; Cell Signaling Technology) for 1 hr at room temperature. Unroofed cells were then washed three times in blocking buffer and incubated with 1:20 diluted 18-nm gold-conjugated goat anti-rabbit secondary antibody (Jackson Laboratories) for 1 hr at room temperature. After rinsing three times in blocking buffer, grids were then washed three times in 150 NaCl, 20 HEPES 7.5 to remove phosphate before staining in 1% uranyl acetate for 1 min. Images were collected on a Tecnai G2 Spirit BioTWIN Transmission Electron Microscope.

## Probability distribution calculation of Piezo1 along the RBC membrane

The Beck RBC model was used to describe the surface of a biconcave disk with diameter 8.0 µm, dimple thickness 1.0 µm, maximum rim thickness 2.5 µm, and area 138 µm$^2$ (*Beck, 1978*). The mean curvature at each point on this surface was calculated as $H\left(x\right) = \frac{1}{2}\left(c_1 + c_2\right)$, where $c_1$ and $c_2$ are the principal curvatures, which are functions of the position $x$ on the surface (*Carmo, 2016*). The following calculation was carried out to assign a (relative) membrane energy, and associated probability density, to Piezo1 at each position $x$ on the RBC surface. The aim of this calculation is not to provide a detailed description of RBC membrane shape but, rather, to test whether Piezo1 localization can couple to curvatures on the RBC surface that are 1–2 orders of magnitude smaller than the Piezo dome curvature. In particular, Piezo1, modeled as a spherical cap (dome) with area 450 nm$^2$ and radius of curvature 42 nm (*Haselwandter et al., 2022a*), was placed into a vesicle whose radius of curvature, if it were spherical, equals $1/H\left(x\right)$. We note that the above conditions specify the area of the vesicle, which equals the free membrane area plus 450 nm$^2$. For positions on the RBC surface at which the orientation (sign) of the RBC curvature matches the orientation of the Piezo1 intrinsic curvature (concave side facing outside the cell) Piezo1 was placed into the vesicle with an inside-out orientation, that is,

matching the orientation of the vesicle curvature. Otherwise, Piezo1 was placed into the vesicle with an outside-out orientation. The energy of the vesicle free membrane (outside the perimeter of the Piezo dome) was calculated through minimization of the Helfrich membrane bending energy

$$G_M = \frac{K_b}{2} \int dA \ (c_1 + c_2)^2 ,$$

where the lipid bilayer bending modulus $K_b \approx 20 \, k_B T$ and the integral is carried out over the entire vesiclefree membrane, subject to the boundary conditions at the Piezo dome-free membrane interface mandated by the shape of the Piezo dome (*Helfrich, 1973*; *Haselwandter et al., 2022b*; *Haselwandter et al., 2022a*). The resulting energy as a function of RBC curvature (and, thus, as a function of the position $x$ on the RBC surface) was applied to the Boltzmann distribution equation to calculate the probability per unit area of observing Piezo1 across the RBC surface. This probability density is graphed on the surface of the Beck RBC model in *Figure 3*. Three major assumptions underlie our calculation of the curvature coupling energy. First, Piezo1 is treated as a spherical cap. Second, for each position on the RBC surface the two principal curvatures, $c_1$ and $c_2$ , are used to define a radius of curvature equal to the reciprocal of the mean curvature. Both the first and second assumption simplify the problem by imposing radial symmetry on the system of the Piezo dome and its surrounding membrane. The third assumption is that Piezo1 is sensing the curvature associated with the overall shape of the RBC, rather than the curvature of a locally specified surface (small membrane compartment) with nearby fixed boundaries.

We also carried out a joint energy minimization of the free membrane as described above while simultaneously permitting Piezo1 to flex, that is, to change its radius of curvature, as part of the energy minimization. This calculation was made possible through our recently determined bending modulus of the Piezo dome itself (*Haselwandter et al., 2022a*). This calculation leads to the conclusion that Piezo1 in an unperturbed RBC should change its mean dome radius of curvature by less than approximately 2 nm as a function of position on the RBC surface.

## Single particle tracking and diffusion analysis

Washed RBCs were adhered to poly-lysine coated 35 mm glass-bottom dishes (MatTeK) and blocked in blocking buffer before incubation with 1:500 dilution of rabbit anti-HA primary (3725; Cell Signaling Technology) for 1 hr at room temperature. Cells were then washed in blocking buffer and incubated with 1:20 diluted 40-nm gold-conjugated goat anti-rabbit secondary antibody (Jackson Laboratories) for 1 hr at room temperature. Cells were washed in blocking buffer and then DPBS before imaging. The movement of colloidal gold particles was observed at room temperature using DIC microscopy on an eclipse Ti2 inverted microscope (Nikon) using a ×40/0.65 objective lens and coupled to an Orca Fusion CMOS camera (Hamamatsu). Video sequences were recorded on Nikon NIS-Elements AR5.4.1 software.

Detection and tracking were carried out in ImageJ imaging analysis software using the TrackMate plugin (*Tinevez et al., 2017*; *Ershov et al., 2021*). First, trainable WEKA segmentation (*Arganda-Carreras et al., 2017*) was employed to generate a model of gold nanoparticles that could be used for spot detection. Detection was then carried out in TrackMate using the Linear Assignment Problem (LAP) tracker. Tracks were exported as *XYT* coordinates and further analyzed in MATLAB. The MATLAB class msdanalyzer (*Tarantino et al., 2014*) was used for MSD analysis of trajectories and estimation of diffusion coefficients.

## Spatial proximity analysis by OTC

Co-immunostained STED images were analyzed using an open-source R code package (*Tameling et al., 2021*; *Tameling and Naas, 2021*) based on OTC. Sections of 64 × 64 pixels (800 × 800 nm) were picked only if they contained fluorescent signal and were analyzed in parallel to get 95% confidence bands.

## Acknowledgements

Super-resolution microscopy work was performed in the Bio-Imaging Resource Center at Rockefeller University, RRID:SCR_017791. The OMX 3D-SIM system was funded by Award Number S10RR031855 from the National Center for Research Resources. This work was also supported

at USC by NSF Grant No. DMR-2051681 and by NSF Grant No. DMR-1554716 (to CAH) and at Rockefeller University by NIH grant GM43949 (to RM). RM is an investigator of the Howard Hughes Medical Institute.

## Additional information

### Funding

| Funder | Grant reference number | Author |
| --- | --- | --- |
| National Center for Research Resources | S10RR031855 | Priyam Banerjee<br>Alison J North |
| National Science Foundation | DMR-2051681 | Christoph A Haselwandter |
| National Science Foundation | DMR-1554716 | Christoph A Haselwandter |
| National Institutes of Health | GM043949 | Roderick MacKinnon |

The funders had no role in study design, data collection, and interpretation, or the decision to submit the work for publication.

### Author contributions

George Vaisey, Conceptualization, Data curation, Formal analysis, Validation, Investigation, Visualization, Methodology, Writing – original draft, Project administration, Writing – review and editing; Priyam Banerjee, Resources, Methodology, Writing – review and editing, Design of imaging experiments; Alison J North, Resources, Methodology, Writing – review and editing, Design of imaging experiments; Christoph A Haselwandter, Formal analysis, Writing – review and editing, Developed calculations of probability distribution along RBC membrane; Roderick MacKinnon, Conceptualization, Formal analysis, Supervision, Funding acquisition, Writing – original draft, Writing – review and editing, Developed calculations of probability distribution along RBC membrane

### Author ORCIDs

George Vaisey ⓘ http://orcid.org/0000-0002-8359-1314
Christoph A Haselwandter ⓘ http://orcid.org/0000-0002-5012-5640
Roderick MacKinnon ⓘ http://orcid.org/0000-0001-7605-4679

### Ethics

All animal procedures were reviewed and approved by the Institutional Animal Care and Use Committee at The Rockefeller University, protocol # 22056.

### Decision letter and Author response

Decision letter https://doi.org/10.7554/eLife.82621.sa1
Author response https://doi.org/10.7554/eLife.82621.sa2

## Additional files

### Supplementary files

• Transparent reporting form

### Data availability

Data generated and analyzed are included in main figures and figure supplements of manuscript. Source files for raw data used for tracking analysis in Figure 4 and Figure 4—figure supplement 1 have been deposited to Dryad: https://doi.org/10.5061/dryad.c2fqz61c7. Source files for raw data used for optimal transport colocalization analysis in Figure 5 have been deposited to Dryad: https://doi.org/10.5061/dryad.1ns1rn8x6.

The following datasets were generated:

| Author(s) | Year | Dataset title | Dataset URL | Database and Identifier |
|---|---|---|---|---|
| Mackinnon R | 2022 | Data from: Piezo1 as a force-through-membrane sensor in red blood cells | http://doi.org/10.5061/dryad.c2fqz61c7 | Dryad Digital Repository, 10.5061/dryad.c2fqz61c7 |
| Mackinnon R | 2022 | Data from: Piezo1 as a force-through-membrane sensor in red blood cells | http://doi.org/10.5061/dryad.1ns1rn8x6 | Dryad Digital Repository, 10.5061/dryad.1ns1rn8x6 |

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
