## [Editor Report]

This study reveals the super-resolution localization of native Piezo1 channels in red blood cells. The data provide further evidence to support the force-through membrane mechanism of mechanotransduction. Notably, the authors find that the distribution of proteins on a cell surface can be governed by membrane curvature.

---

## [Decision Letter]

**Decision letter after peer review:**

Thank you for submitting your article "Piezo1 as a force-through-membrane sensor in red blood cells." for consideration by *eLife*. Your article has been reviewed by 3 peer reviewers, and the evaluation has been overseen by a Reviewing Editor and Kenton Swartz as the Senior Editor. The following individual involved in the review of your submission has agreed to reveal their identity: Zheng Shi (Reviewer #2).

The reviewers all agree this is a strong, interesting, and technically innovative study. They have discussed their reviews with one another, and the Reviewing Editor has drafted this to help you prepare a revised submission. Note that there are several places where new data, analyses, and discussion are requested to help clarify and strengthen to work.

Essential revisions:

1. Please analyze diffusion in rim vs dimple from the data collected already and only need to analyze (see comments from reviewer #1 for detail).

2. Measuring the mesh size would provide essential information. This could be accomplished by segmenting the existing actin super-resolution data to see if the "holes" in the cortex are different along the edges, flat areas, or dimples. The 3D data should be sufficient to see if there are "gross" changes in the actin cytoskeleton in different parts of the cell (see comments from reviewer #1 for detail).

3. The echinocyte data needs to be quantified. (All reviewers raised this as essential)

4. Please discuss these physiological effects/functions in the discussion more to spur the future.

5. More detailed methods for the modeling need to be provided.

6. If possible, please measure membrane density (see reviewer #2 for details).

7. In Figure 2, the claim that "we can often observe multiple triple labels Piezo1 channels" and that "fluorescent spots do not represent multiple Piezo1 channels" needs to be supported by statistical data (see reviewer #3 comments for details).

*Reviewer #1 (Recommendations for the authors):*

I have several suggestions/comments that might improve the manuscript.

1. If the preference of piezo1 to the dimple of the RBC is solely based on an energetic cost to being located in positivity curved membranes, then there should be an "apparent" driving force directing the channel to negatively curved membranes. Thus, I would expect the diffusion coefficient to be different in the dimple vs the edge regions. Did the authors detect different diffusion coefficients in different regions of the cell? This could explain the confinement in these areas.

2. Similar to the above comment, does the size of the actin/spectrin mesh vary in different areas of the cell? Specifically, if the size of the mesh is smaller in the dimple, confinement that is not directly dependent on membrane curvature per se could result. Can the authors show that the mesh size is consistent across the cell's differently-curved sections? Could modeling be used to test these ideas?

3. It would be helpful to quantitate the echinocyte membrane supplemental data and not just show example images. I would also suggest moving this to a main figure.

*Reviewer #2 (Recommendations for the authors):*

My main suggestions are:

1. The Method section describes the use of the Helfrich bending energy of the membrane to understand the distribution of Piezo1 on the RBC surface. However, it is unclear how the modeling was carried out to compare with the experimental observations, potentially due to the referenced PNAS submission. Particularly, it will be helpful to have a reference state (e.g. Piezo1 on a flat membrane) when discussing the curvature-dependent distribution of Piezo1. This is important for the finding to be of general relevance, as most mammalian cells do not take on the special shape of RBCs.

According to the numbers provided in the manuscript, it appears intuitive to think that the ~2-fold difference in Piezo1 density between the dimple and rim of RBCs is due to a depletion effect from the rim rather than an enrichment effect on the dimple.

2. The authors took advantage of the natural bi-concave shape of RBCs to investigate the curvature-mediated distribution of Piezo1. This is quite clever, however, a missing control experiment would be to see if the curvature preference of Piezo1 diminishes upon osmotically swelling the RBCs.

3. Cell membranes often have folds that are below optical resolution. Thus, it is important to compare the density of Piezo1 to that of a lipid membrane marker, to make sure that the observed enrichment of Piezo1 on dimples is not due to higher projected membrane density in these regions.

4. Many membrane proteins (such as integrin) are glycosylated. Are Piezo1 glycosylated? If so, how would the enzymatic deglycosylation treatments affect the interpretation of the lack of clusters?

5. The authors used co-immunostaining to show that 'Piezo1 is not bound to the actin-spectrin meshwork'. However, as Piezo1 is tagged with HA, a more direct experiment seems to be co-immunoprecipitation analysis with Piezo1.

6. A recent publication by Dumitru et al. (Nano Letter 2021) investigated the distribution of Piezo1 in RBCs and Piezo1's interaction with the cytoskeleton. Some of the results are in apparent contradiction with the current manuscript. It will be important to discuss the findings of Dumitru et al.

7. I'd like to draw the authors' attention to a recent preprint (doi: https://doi.org/10.1101/2022.06.22.497259) that quantitatively studies the role of membrane curvature in regulating the distribution of Piezo1 in mammalian cells.

Reviewer #3 (Recommendations for the authors):

1. Regarding 2D STED images (Figure 2) the authors claim that "we can often observe multiple triple labels Piezo1 channels" and that "fluorescent spots do not represent multiple Piezo1 channels", but none of these claims is supported by statistical data. The authors should calculate fluorescence intensity histograms of ROIs surrounding fluorescent spots. This is absolutely necessary to conclude that (a) labeling is near saturation, and that (b) most spots are one single independent channel.

2. Similarly, data related to the localization of Piezo1 channels in echinocytes (Supplementary Figure 6) are not analyzed, but the authors claim "exclusion of Piezo1 channels from protrusions". The authors should quantify the mean curvature of the membrane embedding each Piezo1 channel, then calculate the distribution of channels as a function of curvature (in a histogram), and statistically test if this distribution is biased. This is absolutely necessary to support the claim that Piezo1 does not localize in regions of mismatched, positive membrane curvature.

3. The measurements of nearest neighbor distances in unroofed cells may be skewed by edge effects: Piezo1 channels near the edges have fewer neighbors thus artificially increasing the next neighbor stances. The authors should consider correcting for this effect, perhaps by excluding spots near the edges, or mentioning and discussing this potential artifact.

[Editors’ note: further revisions were suggested prior to acceptance, as described below.]

Thank you for resubmitting your work entitled "Piezo1 as a force-through-membrane sensor in red blood cells." for further consideration by *eLife*. Your revised article has been evaluated by Kenton Swartz (Senior Editor) and a Reviewing Editor.

The manuscript has been improved but there are some remaining issues that need to be addressed, as outlined below:

The reviewers are largely satisfied with your responses to their comments but have asked for a few clarifications to be added to the final publication. Specifically, they ask if you would please better explain how "a comparable diffusion coefficient" was reached, to please add statistical analysis/error bars to the new analyses and to please mention these new data in the main manuscript. They also ask you to provide the reference for the statement that membrane is "smooth" in red blood cells.

*Reviewer #1 (Recommendations for the authors):*

The authors have addressed the majority of my concerns in this revision.

*Reviewer #2 (Recommendations for the authors):*

1. In the additional diffusion analysis, it's unclear how "a comparable diffusion coefficient" was reached, please add proper statistical analysis or error bars. Also, why not mention this new analysis in the manuscript?

2. Response to the analysis of echinocyte membrane is not fully satisfactory. Can the authors at least segment out regions of obvious protrusion and show that the density of Piezo1 is significantly lower in these regions?

3. Response to the request regarding modeling. The reviewer will need significantly more time to digest the newly published PNAS papers, therefore cannot comment on the completeness of the response in a timely manner.

4. Response to controlling for membrane density. Please provide reference to support the claim that membrane is "smooth" in red blood cells.

Membrane folds and fluctuations below resolution limit will lead to a difference between the real membrane area and the projected membrane area. The analysis and comparison of Piezo1 density could be affect, especially if the rim and the dimple of RBC have different amount of such membrane fluctuations.

*Reviewer #3 (Recommendations for the authors):*

All my concerns have been addressed. This is now a very nice manuscript.

---

## [Author Response]

Essential revisions:1. Please analyze diffusion in rim vs dimple from the data collected already and only need to analyze (see comments from reviewer #1 for detail).

Regarding the diffusion coefficient, as we show in author response image 1, the diffusion coefficient is not different in the dimple versus the rim. We wish to emphasize that on theoretical grounds one should not necessarily expect a difference. If binding sites for Piezo existed in one region and not another, then diffusion would be slowed and Piezo would be concentrated in that region. If on the other hand barriers to diffusion existed in one region, this would slow diffusion without concentrating Piezo. According to the curvature coupling model, Piezo is more stable in the dimple and thus is attracted there, but the diffusion coefficient is determined by very shortrange (compared to the size of the dimple), short timescale (compared to the time spent in the dimple) interactions. Thus, we view this process more like one modelled by diffusion of a particle in potential energy field with a constant diffusion coefficient.

For completeness, we analyzed our gold-tracking images picking 3 second durations where the gold could be easily identified as in the dimple or the rim of the RBC membrane. Mean squared displacement analysis of n=6 (dimple) and n=8 (rim) tracks shows a comparable diffusion coefficient, consistent with the average calculated in Figure 4 E of ~0.0075 µm^2^s^-1^ (black is averaged MSD and red is the straight line fit with an R^2^ of 0.98 for dimple measurements and 0.94 for rim measurements).

**Author response image 1. sa2fig1:** 

2. Measuring the mesh size would provide essential information. This could be accomplished by segmenting the existing actin super-resolution data to see if the "holes" in the cortex are different along the edges, flat areas, or dimples. The 3D data should be sufficient to see if there are "gross" changes in the actin cytoskeleton in different parts of the cell (see comments from reviewer #1 for detail).

We have addressed this point by segmenting actin fluorescence at the dimple and the rim as we did for Piezo1. Then, using Imaris Spot detector we estimated the relative surface density of actin signal at these two regions. We find no statistical enrichment of actin signal at the dimple, indicating that the actin-spectrin meshwork is unlikely to be denser at the dimple. We have appended this data to Figure 3C. This result is consistent with not finding a significant enrichment of other membrane proteins, Band3 and KCNN4, which bind to the cytoskeleton, at the dimple.

3. The echinocyte data needs to be quantified. (All reviewers raised this as essential)

Thank you for this suggestion. We attempted to analyze Piezo1 distribution as a function of curvature along echinocyte membranes but were not successful in doing so rigorously enough. The major problem is that the protrusions, on the order of 100-500nm are below or around the axial resolution limit in our 3D-SIM analysis and so determining the radius of curvature is inaccurate here and varies depending on how you contour the actin surface signal in Imaris. Any estimation of Piezo1 distribution as a function of membrane curvature from these analyses would thus be inappropriate. Instead, we are leaving Figure 3-figure suppelement 3 as a qualitative demonstration that fluorescent Piezo1 spots appear depleted from membrane protrusions in echinocytes and we state this clearly in the text.

4. Please discuss these physiological effects/functions in the discussion more to spur the future.

– On the non-clustered distribution of Piezo1 in the context of red blood cell force sensation: “A relatively homogenous membrane distribution of Piezo1 might be well suited to RBCs that are tumbling through solution, experiencing shear forces from all angles, and may contribute to the remarkable plasticity of RBC shapes under stress (Dupire, Socol, and Viallat 2012).”

– On curvature coupling of Piezo1 extending beyond red blood cells: “Whilst the curved biconcave shape of RBCs is unique, the good agreement between our experimental data and theoretical prediction based on energetic curvature coupling would suggest an inherent relationship between Piezo’s intrinsic curvature and the surrounding membrane that is broadly relevant to other cell types. This is because the curvature coupling theory, which was developed with experiments on lipid bilayer vesicles, depends only on the bending elasticity of membranes and the intrinsic curvature of

Piezo1, not on the unique shape of RBCs. It will be interesting to explore whether Piezo1, for example, is enriched in membrane invaginations or other curvature-matched membrane regions in different cell types.”

– On the functional relationship between Piezo1 and Gardos channel activity not requiring close spatial proximity in red blood cells: “Additionally, we asked whether Piezo1 might be co-localized with the Ca^2+^-activated K^+^ (Gardos) channel. Ca^2+^ flux into the RBC through Piezo1 and subsequent activation of the Gardos channel is thought to be the initial step in mechanically activated RBC volume regulation (Danielczok et al. 2017; Cahalan et al. 2015). We do not, however, observe significant spatial proximity between Piezo1 and Gardos channels. Given the small size of RBCs, approximately 1 µm thick in the dimple and 2 – 3 µm thick in the rim, and the Piezo1 density of around 0.5 per µm^2^, it seems that co-localization is probably not necessary for rapid opening of the Gardos channel following Piezo activation. For diffusion in 3 dimensions, the mean dispersion time can be approximated by τ=x26D,x the mean distance and *D* the diffusion coefficient. The apparent diffusion coefficient for intracellular Ca^2+^ is in the range 13 – 65 µm^2^*/s* (Nakatani, Chen, and Koutalos 2002), meaning that Ca^2+^ should equilibrate within tens of ms within a RBC”

– On Piezo1 mechanosensation in red blood cells: “However, that Piezo1 distributes on the surface of RBCs by sensing the membrane curvature, by corollary, means that it must have the capacity to sense lateral membrane tension by a force-throughmembrane mechanism. To see this, consider that Piezo1 has been shown to flex, i.e., change its curvature, as a function of local membrane curvature (Haselwandter et al., 2022). Consider also that membrane tension changes membrane curvature by flattening it. In this manner, curvature sensing and lateral membrane tension sensing are inextricably linked. This linkage lies at the heart of the membrane dome model of Piezo gating (Guo and MacKinnon 2017; Haselwandter and MacKinnon 2018; Haselwandter et al., 2022). When RBCs squeeze through small capillaries, the membrane shape as well as the transmembrane pressure change transiently. The resulting changes in membrane curvature and lateral membrane tension can both, according to the membrane dome model of Piezo gating, open Piezo1’s pore to trigger Ca^2+^ entry”

We believe we have discussed our observed experimental data about Piezo1 distribution and diffusion properties in red blood cells in broader contexts of both Piezo1 force sensation in physiology as well as red blood cell mechanosensation. Any additional discussion would be pure speculation and outside the scope of this work.

5. More detailed methods for the modeling need to be provided.

Thank you for this suggestion. We understand why the methods seemed incomplete because they referred to papers that were not yet published. Now the methods are discussed in detail in two recently published papers (now referenced in the paper): Haselwandter, C. A., Guo, Y. R., Fu, Z., and MacKinnon, R. (2022). Quantitative prediction and measurement of Piezo's membrane footprint. Proc. Natl. Acad. Sci. U.S.A., 119(40), e2208027119, https://www.pnas.org/doi/abs/10.1073/pnas.2208027119; Haselwandter, C. A., Guo, Y. R., Fu, Z., and MacKinnon, R. (2022). Elastic properties and shape of the Piezo dome underlying its mechanosensory function. Proc. Natl. Acad. Sci. U.S.A., 119(40), e2208034119, https://www.pnas.org/doi/10.1073/pnas.2208034119. We feel that repeating the detailed methods here would potentially confuse the reader. Instead, we focus on how this previous work connects to the current manuscript, and discuss in detail all the parameter values needed to apply this model to the experiments described here. We have updated the citations of these two previous papers to their final form.

6. If possible, please measure membrane density (see reviewer #2 for details).

We disagree with Reviewer #2 comment 3. There is an abundance of scanning electron microscopy of red blood cells that show, unlike most other types of cells, that the membrane is relatively smooth across the cell and does not show local “folds” or calveolae. If there are nanoscale folds, they are beyond detection limits by our system and a membrane fluorescent dye would not change this.

7. In Figure 2, the claim that "we can often observe multiple triple labels Piezo1 channels" and that "fluorescent spots do not represent multiple Piezo1 channels" needs to be supported by statistical data (see reviewer #3 comments for details).

We appreciate the suggestion of statistical analysis of the frequency of observing triple-labelled Piezo1 channels in our images. We were hesitant to do this initially since some fluorescent puncta do not resolve after deconvolution into anything beyond a blurred spot. This is common in STED and is likely due to photobleaching of antibody-conjugated dyes during the image scanning. These unresolved spots could represent single, double, or triple-labelled Piezo1 but since we can’t resolve them, we can’t count them. Still, we did the following semiquantiative analysis: first we thresholded fluorescent Piezo1 spots on their intensity, analyzing only those spots with an intensity above 20,000 units, as measured by line-scan analysis in ImageJ. We then counted how many bright pixels could be resolved after deconvolution to determine if the Piezo1 is single, double or triple labelled. A frequency distribution of these measurements is shown in Figure 2—figure supplement 2, with the x axis labelled as “putative no. of antibodies bound”. Assuming a simple binomial distribution of antibody binding with no cooperative behavior, our data indicate we are at ~60% antibody binding. We could of course be biasing against choosing single-labelled Piezo1 channels which may be more likely to represent the dim fluorescent piezo1 spots but it is worth noting that, assuming a binomial distribution, for us to be underestimating Piezo1 channel number by an order of magnitude, with ~10% antibody binding, we would expect to see approximately zero triple-labelled Piezo1 channels, which is not the case. Even if triplets only made up around 6% of our Piezo1 fluorescent spots we would expect to still have only have ~20% channels unlabelled, thus not markedly changing our channel number per red blood cell estimation.

With regards to our claim that fluorescent spots do not represent multiple Piezo1 channels we have first clarified some points in the text. We never see bright spots resolve into anything more than a triplet, consistent with these spots not representing multiple Piezo1 channels. We make this clearer in the text. Additionally, we edited figure 2 to show line-scanning analysis of adjacent bright pixels (putative antibody binding sites). We find a consistent 25nm measurement in this analysis which we write in the text. This distance is consistent with our gold particle labelling electron microscopy studies.

Reviewer #1 (Recommendations for the authors):I have several suggestions/comments that might improve the manuscript.1. If the preference of piezo1 to the dimple of the RBC is solely based on an energetic cost to being located in positivity curved membranes, then there should be an "apparent" driving force directing the channel to negatively curved membranes. Thus, I would expect the diffusion coefficient to be different in the dimple vs the edge regions. Did the authors detect different diffusion coefficients in different regions of the cell? This could explain the confinement in these areas.

See our response to essential revision 1

2. Similar to the above comment, does the size of the actin/spectrin mesh vary in different areas of the cell? Specifically, if the size of the mesh is smaller in the dimple, confinement that is not directly dependent on membrane curvature per se could result. Can the authors show that the mesh size is consistent across the cell's differently-curved sections? Could modeling be used to test these ideas?

See our response to essential revision 2

3. It would be helpful to quantitate the echinocyte membrane supplemental data and not just show example images. I would also suggest moving this to a main figure.

See our response to essential revision 3

Reviewer #2 (Recommendations for the authors):My main suggestions are:1. The Method section describes the use of the Helfrich bending energy of the membrane to understand the distribution of Piezo1 on the RBC surface. However, it is unclear how the modeling was carried out to compare with the experimental observations, potentially due to the referenced PNAS submission. Particularly, it will be helpful to have a reference state (e.g. Piezo1 on a flat membrane) when discussing the curvature-dependent distribution of Piezo1. This is important for the finding to be of general relevance, as most mammalian cells do not take on the special shape of RBCs.According to the numbers provided in the manuscript, it appears intuitive to think that the ~2-fold difference in Piezo1 density between the dimple and rim of RBCs is due to a depletion effect from the rim rather than an enrichment effect on the dimple.

We have included the references to the now published PNAS studies. These companion papers lay out calculations of parameters (Piezo1 intrinsic curvature) used for our modeling in this study.

2. The authors took advantage of the natural bi-concave shape of RBCs to investigate the curvature-mediated distribution of Piezo1. This is quite clever, however, a missing control experiment would be to see if the curvature preference of Piezo1 diminishes upon osmotically swelling the RBCs.

We disagree that this is an important control experiment. We have already used the natural variability in the biconcave shape of RBCs to show that those cells that are less biconcave show a lower degree of Piezo1 enrichment in the dimple.

3. Cell membranes often have folds that are below optical resolution. Thus, it is important to compare the density of Piezo1 to that of a lipid membrane marker, to make sure that the observed enrichment of Piezo1 on dimples is not due to higher projected membrane density in these regions.

See response to essential revision 6

4. Many membrane proteins (such as integrin) are glycosylated. Are Piezo1 glycosylated? If so, how would the enzymatic deglycosylation treatments affect the interpretation of the lack of clusters?5. The authors used co-immunostaining to show that 'Piezo1 is not bound to the actin-spectrin meshwork'. However, as Piezo1 is tagged with HA, a more direct experiment seems to be co-immunoprecipitation analysis with Piezo1.

We strongly disagree that co-immunoprecipitatoin is a better experiment in this case. Co-IP experiments can frequently lead to false positives especially when pulling down on a low abundant protein (Piezo1) and looking for interaction/non-interaction with a massively more abundant (and sticky) protein like spectrin/actin.

6. A recent publication by Dumitru et al. (Nano Letter 2021) investigated the distribution of Piezo1 in RBCs and Piezo1's interaction with the cytoskeleton. Some of the results are in apparent contradiction with the current manuscript. It will be important to discuss the findings of Dumitru et al.

Thank you for this suggestion, we have included reference to this paper in our discussion however we are hesitant to talk about these findings too much since the authors use a commercial anti-Piezo1 polyclonal antibody and show its specificity by western blot analysis, rather than eg immunostaining of Piezo1 KO red blood cells versus wildtype red blood cells. Analysis of Piezo1 clusters uses diffraction-limited confocal microscopy where indeed the “cluster size” of ~275nm is approximately equal to diffraction-limited xy resolution. Additionally, colocalization with spectrin is assessed by confocal microscopy where resolution is limited. Because of the abundance of spectrin in red blood cells and the small size of these cells overall, we would expect every membrane protein in red blood cells to show significant colocalization with spectrin using this approach.

7. I'd like to draw the authors' attention to a recent preprint (doi: https://doi.org/10.1101/2022.06.22.497259) that quantitatively studies the role of membrane curvature in regulating the distribution of Piezo1 in mammalian cells.

Thank you for drawing our attention to this preprint. We have referenced this study in our discussion as orthogonal evidence for curvature coupling of Piezo1 in cellular membranes. However, we would want to discuss the quantitative modelling of Piezo1’s “nano-geometry” with the authors since we have some serious disagreements with their calculations but do not think our paper is the appropriate forum for this.

Reviewer #3 (Recommendations for the authors):1. Regarding 2D STED images (Figure 2) the authors claim that "we can often observe multiple triple labels Piezo1 channels" and that "fluorescent spots do not represent multiple Piezo1 channels", but none of these claims is supported by statistical data. The authors should calculate fluorescence intensity histograms of ROIs surrounding fluorescent spots. This is absolutely necessary to conclude that (a) labeling is near saturation, and that (b) most spots are one single independent channel.

See our response to essential revision 7

2. Similarly, data related to the localization of Piezo1 channels in echinocytes (Supplementary Figure 6) are not analyzed, but the authors claim "exclusion of Piezo1 channels from protrusions". The authors should quantify the mean curvature of the membrane embedding each Piezo1 channel, then calculate the distribution of channels as a function of curvature (in a histogram), and statistically test if this distribution is biased. This is absolutely necessary to support the claim that Piezo1 does not localize in regions of mismatched, positive membrane curvature.

See our response to essential revision 2

3. The measurements of nearest neighbor distances in unroofed cells may be skewed by edge effects: Piezo1 channels near the edges have fewer neighbors thus artificially increasing the next neighbor stances. The authors should consider correcting for this effect, perhaps by excluding spots near the edges, or mentioning and discussing this potential artifact.

Thank you for this helpful suggestion. We have repeated our Piezo1 spot-spot distance on 2D STED images of unroofed membranes, this time masking out spots that lie on the edges of the membranes. Doing so does not change the mean spot-spot distance significantly, nor alter our interpretation of this data.

[Editors' note: further revisions were suggested prior to acceptance, as described below.]

The reviewers are largely satisfied with your responses to their comments but have asked for a few clarifications to be added to the final publication. Specifically, they ask if you would please better explain how "a comparable diffusion coefficient" was reached, to please add statistical analysis/error bars to the new analyses and to please mention these new data in the main manuscript. They also ask you to provide the reference for the statement that membrane is "smooth" in red blood cells.

We have included the diffusion analysis of the rim versus the dimple in a new figure, figure 4—figure supplement 2. In addition to showing the mean MSD trajectories we have calculated diffusion coefficients from the same n=6 (dimple) and n=8 (rim) individual trajectories and plot these on a scatter plot with the mean and SEM. There is no statistically significant difference between the dimple and rim as analyzed by an upaired t test. This additional analysis is referenced on page 13 of the resubmitted manuscript.

We have included references to two studies analyzing RBCs by electron microscopy on page 11 of the resubmitted manuscript where we make the point that our modeling of curvature coupling between Piezo1 and the membrane curvature of RBCs implies a smooth RBC surface, which is consistent with EM analysis of healthy RBCs.